Research



**Subject Category:**
Biology (whole organism)

**Subject Areas:**
ecology/behaviour/physiology

**Keywords:**
Lithodidae, reproduction, sperm depletion, vasa deferentia, crustacean, vasosomatic index

**Author for correspondence:**
Katrin Pretterebner
e-mail: k.pretterebner@hotmail.com



# Temperature-dependent seminal recovery in the southern king crab *Lithodes santolla*

Katrin Pretterebner[1,2,3], Luis Miguel Pardo[2,3] and Kurt Paschke[3,4]

[1]Facultad de Ciencias, Programa de Doctorado en Biología Marina, and [2]Facultad de Ciencias, Instituto de Ciencias Marinas y Limnológicas, Laboratorio Costero de Calfuco, Universidad Austral de Chile, Valdivia 5090000, Chile
[3]Centro de Investigación de Dinámica de Ecosistemas Marinos de Altas Latitudes (IDEAL), Valdivia 5090000, Chile
[4]Instituto de Acuicultura, Universidad Austral de Chile, Puerto Montt 5480000, Chile

KP, 0000-0003-1150-7475

Male-biased fishery management can provoke depletion of seminal reserves, which is the primary cause of sperm limitation. Therefore, identifying factors which contribute to the vulnerability to depletion of seminal reserves is a priority. The present study aimed to determine the effect of temperature on the recovery rate of sperm and seminal reserves after their depletion in *Lithodes santolla*, an important fishery resource in southern Chile. Sperm and seminal reserves were not fully recovered within 30 days. Temperature significantly affected seminal recovery: after 30 days the recovery index increased to 40% and 21% at 9°C and 12°C, respectively. The twice as fast seminal recovery at 9°C may be explained by the zone of origin of the individuals in this study (northern distributional limit), and 12°C may be close to the threshold of temperature tolerance. *Lithodes santolla* populations subject to intense male-only fisheries may be vulnerable to depletion of seminal reserves and a climate change scenario could additionally aggravate the risk of seminal depletion in *L. santolla* in its northern distributional limit.

## 1. Introduction

Most crustacean fisheries are managed by a male-biased or exclusively male extraction strategy [1]. However, harvesting large males can trigger changes in the mating dynamics of a population, thereby reducing overall reproductive success [2,3]. The population structure can be altered by selectively harvesting

large males due to skewing the operational sex-ratio towards females and decreasing the average size of males [4]. A reduced density of males in the population may affect the frequency of encounters of females with males and cause difficulties for females to find mates [5]. Owing to a lower availability of dominant males, available males potentially mate more frequently than in non-fished populations [6]. Hence, males might deplete their sperm reserves faster than they are able to recover them, and transfer reduced sperm to females, an issue which has been termed sperm depletion [7–11].

Traditionally, sperm has been assumed to be cheap in production [12,13]. However, sperm is usually delivered along with seminal fluids and the production of ejaculates is slow and costly for males [14]. The capacity of male decapods to transfer sperm is limited by the exhaustion of sperm and seminal fluids (i.e. seminal reserves) and their regeneration rate [15]. Therefore, an important factor involved in seminal depletion may be the time period necessary to fully recover seminal reserves after mating. Until now, sperm regeneration rates have been studied in few decapod species. The subtropical blue crab *Callinectes sapidus* requires 9–20 days to recover its vasa deferentia weight after two consecutive matings [7]. By contrast, in the anomuran species *Paralithodes brevipes* and *Hapalogaster dentata*, sperm in the vasa deferentia is not fully recovered even after 28 and 20 days, respectively, after depletion [16,17]. In crab species with a rapid seminal recovery rate, males might be less vulnerable to seminal depletion. Hence, it is fundamental to determine the recovery rate of seminal reserves in individual crab species and identify factors that contribute to the variability in the recovery period.

Crustaceans are ectothermic, meaning that their body temperature depends on the water temperature. Temperature determines the velocity of the energetic metabolism and regulates most biological and behavioural patterns. Environmental factors such as temperature might modulate the male's ability to regenerate seminal material apart from intrinsic ones (e.g. moult state after spermatophore extrusion [18]). So far, the effect of temperature on the production of spermatophores has only been studied in a commercially important penaeid shrimp species in the context of aquaculture research [19]. In male *Penaeus setiferus*, the regeneration process was estimated according to visual evaluation of the spermatophores after electric shock-induced ejaculation (i.e. electroejaculation). Temperature affected the period necessary for regeneration; in this tropical species, spermatophores were replaced faster at 33°C (within 144 h) compared to 25°C (192 h), however, at cost of a reduced sperm quality [19]. Bugnot & López Greco [20] reported that sperm production in the red claw crayfish *Cherax quadricarinatus* was higher between 27°C and 29°C compared to 23°C or 31°C. The effect of temperature on the recovery rate of sperm and seminal reserves in crustaceans has not been evaluated and discussed in the context of seminal depletion triggered by selective fishery.

The southern king crab *Lithodes santolla* (Molina, 1782) constitutes one of the most important fishery resources in the Region of Magallanes and Chilean Antarctica (Chilean 5-year mean landing around 5965 tons [21]). *Lithodes santolla* is distributed in subantarctic and cold-temperate environments from the Beagle Channel (southernmost parts of Argentina and Chile) along the southeastern Pacific to Talcahuano [22,23] and along the southwestern Atlantic in the deeper parts of the continental slope off Uruguay [24]. Chilean fishery of *L. santolla* has been concentrated mainly in Porvenir, Punta Arenas, Puerto Natales and around the island of Chiloé. The extraction of *L. santolla* is regulated by the strategy of 'SSS' (size, sex and season). Harvesting females is prohibited year-round and the fishery is closed in the Region of Magallanes and Chilean Antarctica from 1 December until 30 June and in the northern regions (X, XIV and XI) from 1 December until 31 January. The minimum harvest size (carapace length (CL)) for *L. santolla* depends on the location: 120 mm south and 100 mm north of 46° S.

The annual reproductive cycle of *L. santolla* in the Beagle Channel starts in late November to early December and mating pairs can be found in the population for approximately one month [25]. Precopulatory mate-guarding and mating occur between an old-shelled male and a female that has recently moulted. The male reproductive system of *L. santolla* consists of paired testes, secretory canals and vasa deferentia [26]. In the testes, development occurs from the spermatogonia to spermatozoid which are packaged in spermatophores and stored in the vasa deferentia [27]. The vasa deferentia are further connected to the fifth pereiopods where, through apertures located at the coxae, the spermatophores are extruded during copulation [26]. External fertilization takes place immediately after oviposition within the brood chamber formed by the pleon flapped below the cephalothorax. Females of *L. santolla* brood the embryos that are attached to their pleopods for approximately 9–10 months and larvae hatch between mid-September and October [28]. Fecundity (number of eggs per brood) increases with female size and *L. santolla* produces between 5500 and 60 000 eggs per clutch [29]. Even though fundamental knowledge on this species exists [30], studies focusing on male reproductive aspects are scarce [27]. Moreover, biological information of *L. santolla* is largely restricted to the southern limit of distribution (Beagle Channel) without data describing reproductive aspects of

males derived from their northern distributional limit. Intraspecific latitudinal variability in the reproductive pattern and traits has been documented in crustacean species [31–34]. Consequently, *L. santolla* in its northern distributional limit (this study) is exposed to different prevailing environmental conditions (e.g. temperature, photoperiod) which probably have led to variations in the reproductive pattern between the southern and northern limits of its distribution (i.e. extended window of reproduction in northern distributional limit).

As exploitation of this lithodid is regulated by a large male-only management strategy, the target species may be susceptible to depletion of seminal reserves which makes it a suitable model species to identify factors that contribute to the vulnerability to seminal depletion. This is important for fisheries and it is crucial to obtain a greater understanding of males' reproductive biology in *L. santolla*. As *L. santolla* inhabits cold seawater environments and has a low energy metabolism, a slow sperm and seminal recovery rate after depletion of seminal reserves is expected. The objective of this study was to determine the effect of temperature on the recovery rate of sperm and seminal reserves after their depletion in *L. santolla* from its northern distributional limit.

## 2. Material and methods

Physiologically mature similar-sized males of *L. santolla* (physiological maturity in the Beagle Channel greater than 75 mm CL, see [35]) were collected in October 2016 (caught with commercial traps) from the Seno de Reloncaví, X region, Chile (41°45′47.1″ S; 73°05′20.1″ W) which corresponds to the northern limit of distribution of this species. Individuals were transported to the Laboratorio de Ecofisiología de Crustáceos (LECOFIC) of the Universidad Austral de Chile in Puerto Montt. Crabs were acclimated in the laboratory for approximately one month prior to the start of the experiment with flowing seawater, air supply and ad libitum food (*Mytilus chilensis*). Both temperature conditions (9°C and 12°C) were conducted simultaneously from mid-November to mid-December 2016.

To simulate depletion of seminal reserves and to ensure standardization of initial condition of males, individuals of *L. santolla* were stimulated for electroejaculation repetitively on four successive days through short electric shocks of 12 V AC [36]. Electrodes were placed ventrally on the soft section of the opened pleon (figure 1*a*). After seminal depletion (i.e. no ejaculation after electrostimulation), individuals were maintained in two tanks at 9°C and 12°C each with flowing seawater (32 psu), air supply ($O_2$ saturation was guaranteed through bubbling and movement of the water by pumps) and ad libitum food. The first experimental temperature of $9.2 \pm 0.09$°C (mean ± s.e.) was provided by a chiller. The second experimental temperature of $12.04 \pm 0.06$°C (mean ± s.e.) was maintained using submersible aquarium heaters with digital controllers. These two experimental temperatures are referred to as 9°C and 12°C in the text, reflecting the mean sea surface temperatures in the study area, Seno de Reloncaví, during the austral winter (May–August, 9°C) and throughout the year (12°C) [37]. Individuals in the lower temperature condition were acclimatized for 2–3 days through a gradual temperature decrease until reaching 9°C. Temperature was measured every day with a digital thermometer (WTW Cond 330i with sensor WTW TetraCon 325; precision of 0.1°C). To identify crabs individually, they were marked with numbered cable ties.

To determine the effect of temperature on the recovery rate of sperm and seminal reserves, individuals were sacrificed (thermal shock −80°C for 15 minutes) after 0, 15 and 30 days. Crabs without electroejaculation were used as controls and sacrificed after 0 and 30 days. After the corresponding experimental period, both vasa deferentia and the hepatopancreas were dissected (figure 1*b*). The left vasa deferentia were used to estimate their dry weight (seminal material) and to calculate the vasosomatic index (VSI, $n = 4$). The VSI has been suggested as an indicator of the reproductive condition of males [8,15,38]. The VSI (expressed as a percentage) was calculated: $\mathrm{VSI} = (2 \times \mathrm{VDW}/\mathrm{BDW}) \times 100$, with VDW being the dry weight of the left vasa deferentia (oven dried for 5 days at 70°C and weighed to a precision of 0.0001 g) and BDW being the dry weight of the complete body (oven dried for 5 days at 70°C and weighed to a precision of 0.01 g). The weight of the left vasa deferentia was doubled to calculate the VSI of the whole male reproductive system. For the estimation of the VSI, dry weight of crabs without walking legs and chelae was used to increase accuracy.

The right vasa deferentia were used to examine histological cross-sections ($n = 4$ in each initial group, $n = 5$ in each recovery group, $n = 3$ after 30 days at 12°C) and to calculate the area covered by sperm (stored in spermatophores) in the vasa deferentia lumen. To prepare histological cross-sections, the right vasa deferentia were fixed in Bouin solution for at least 2 days. Then, samples were sequentially passed through a 50–70–80–96–100% ethanol series for 30 min each, 100% ethanol–butylic alcohol

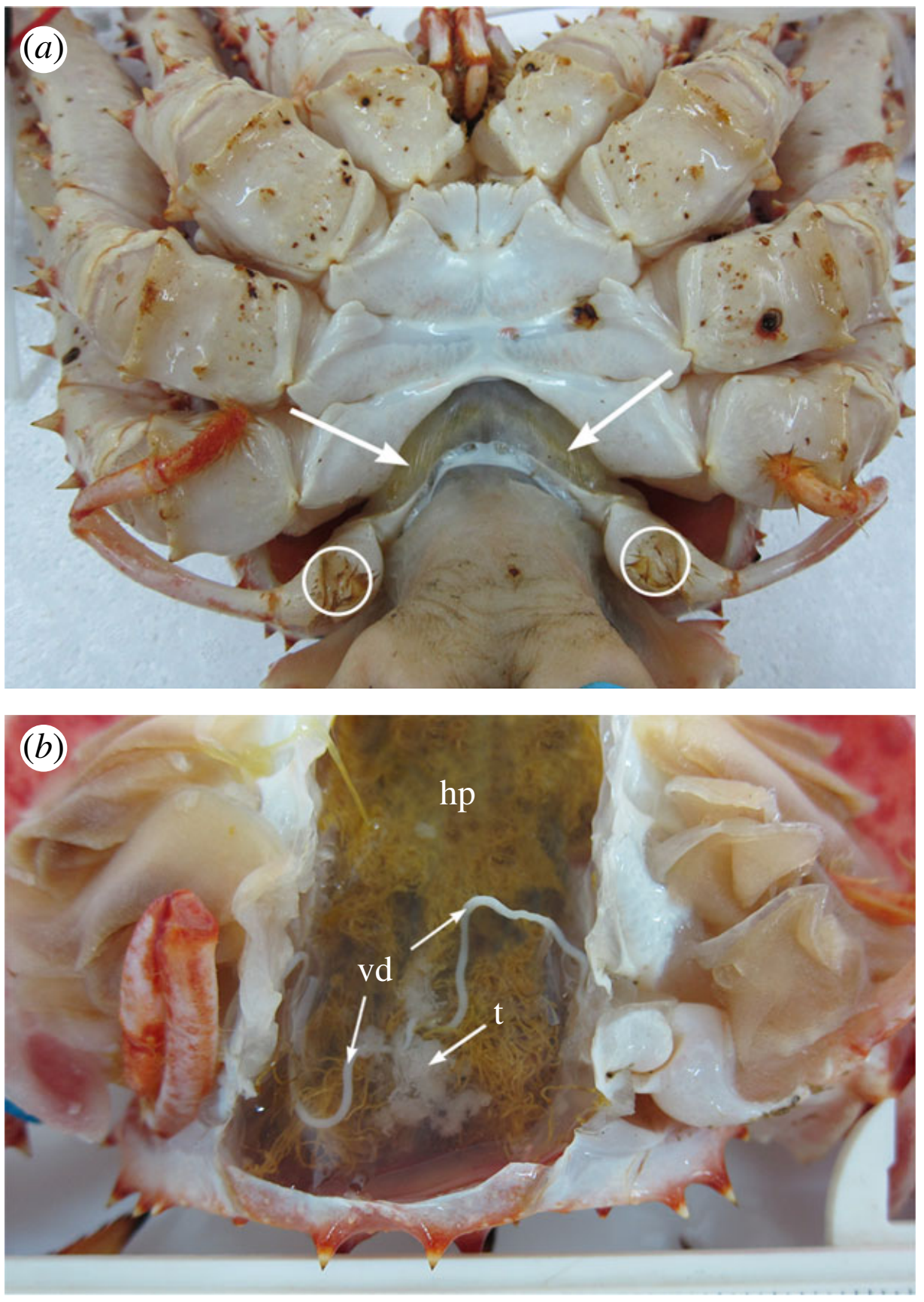


**Figure 1.** Experimental procedure of *Lithodes santolla*. (*a*) Ventral view showing the opened abdominal flap (pleon). Placement of electrodes during electroejaculation on the inner section of the opened abdominal flap (arrows) and gonopores (openings at the coxae of the fifth pereiopods) where ejaculate is extruded (circles) are indicated. (*b*) Dissection of paired vasa deferentia (vd), testes (t) and hepatopancreas (hp).

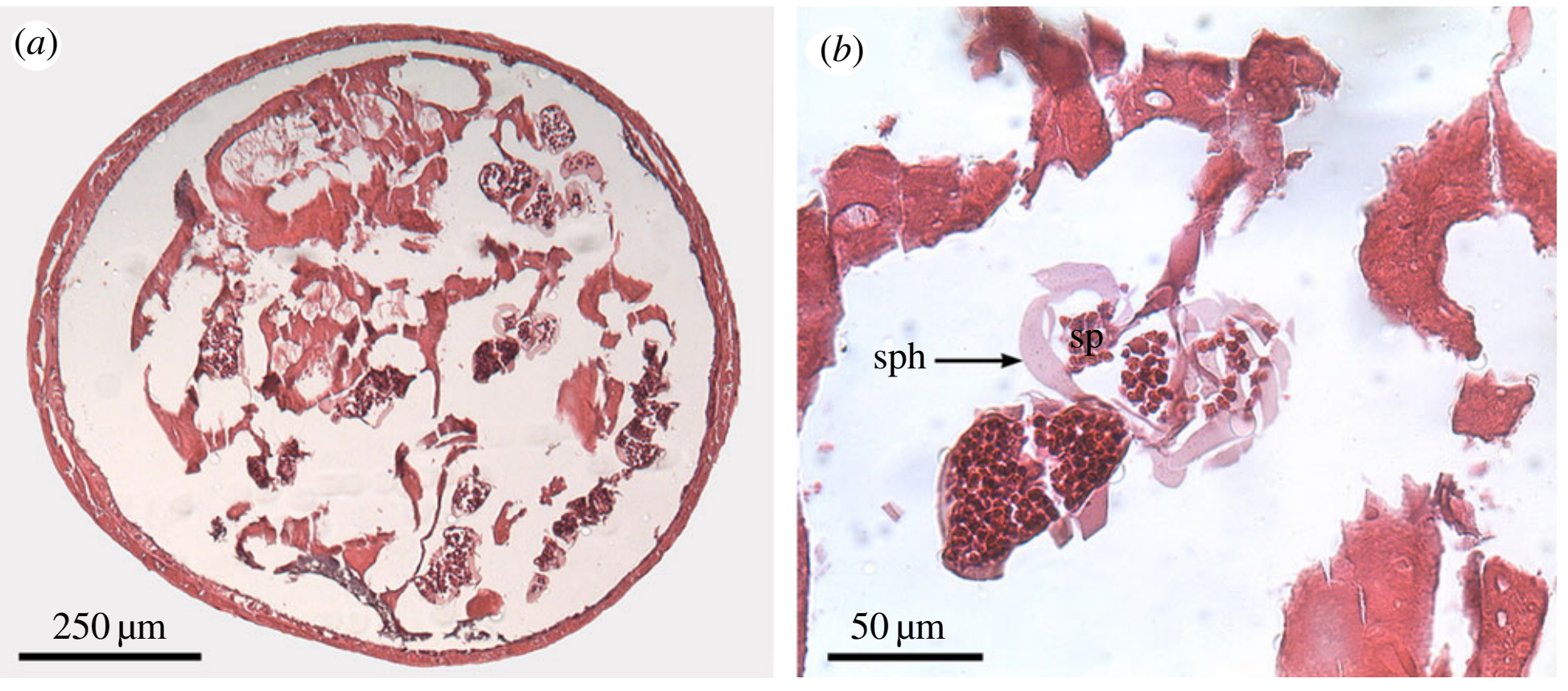


**Figure 2.** Histological cross-section of initial vasa deferentia of *Lithodes santolla* without electroejaculation. (*a*) Overview of section used to estimate area covered by sperm in the lumen. (*b*) Detailed view displaying spermatophores (sph) filled with sperm (sp).

(1 : 1 v/v) for 30 min and butylic alcohol for 25 min (twice). Samples were embedded in paraffin and serial sections of 6 µm were cut and stained with haematoxylin-eosin. One slice of the middle section of the vasa deferentia, which corresponds to the section where spermatophore reserves are located [35], was used to estimate area covered by sperm in the lumen (figure 2*a*,*b*). Photos were taken with a digital camera (Q Imaging Go-3) of histological cross-sections under a light microscope (OLYMPUS BX51) at 10× magnification for further analysis. Photos were analysed using the image processing software ImageJ 1.52a [39]. The area covered by sperm was identified through adjusting the brightness of the colour threshold and measured. The sperm area was then doubled to calculate the

sperm coverage of the paired vasa deferentia. Sperm area is presented in relation to CL and multiplied by 1000.

The hepatopancreas was extracted and the dry weight was determined ($n = 5$ in each group). The hepatosomatic index (HSI) was calculated: HSI = (HDW/BDW) × 100. HDW was the dry weight of the hepatopancreas and BDW was the dry weight of the animal without legs. In crustaceans, the hepatopancreas is an organ involved in digestive processes and it is important in the assimilation of energy, storage and mobilization of resources during the moulting process. Thus, the size and weight of the hepatopancreas provide a good indication of somatic resources [40].

## 2.1. Data analyses and statistics

We established a recovery index (RI), which refers to the recovery rate of seminal material expressed as the percentage change in dry weight of the vasa deferentia between initially electroejaculating individuals ($n = 5$) and after the corresponding experimental period ($n = 5$ each) in relation to dry weight of the vasa deferentia of the control (i.e. without electroejaculation, $n = 4$). The dry weight of the vasa deferentia remaining after electroejaculation was not considered. To assess the net change rate of VDW of the control animals (i.e. natural fluctuation of the vasa deferentia without electroejaculation), change rate (in %) of vasa deferentia dry weight after 30 days ($n = 5$ each and $n = 4$ in initial group) in relation to the initial vasa deferentia dry weight ($n = 4$) was calculated for 9°C and 12°C.

The following parameters were used in the RI in electroejaculated individuals and the net change rate of VDW in the control animals. VDW = vasa deferentia dry weight; E = crab electroejaculated at beginning of experiment; C = control (i.e. without electroejaculation); $t_0$ = at beginning of the experiment; and $t_1$ = after corresponding time period of either 15 or 30 days.

$$A = \mathrm{VDW}_{\mathrm{E}t_1} - \mathrm{VDW}_{\mathrm{E}t_0} = \text{increment in weight of seminal material after } t_1,$$
$$B = \mathrm{VDW}_{\mathrm{C}t_0} - \mathrm{VDW}_{\mathrm{E}t_0} = \text{quantity of seminal material to be recovered,}$$
$$\mathrm{RI}\ (\%) = \frac{A}{B} \times 100$$
$$\text{and}\quad \text{change rate VDW}\ (\%) = \frac{\mathrm{VDW}_{\mathrm{C}t_1} - \mathrm{VDW}_{\mathrm{C}t_0}}{\mathrm{VDW}_{\mathrm{C}t_0}} \times 100.$$

Both RIs and net change rates of VDW in control individuals were estimated through bootstrapping of the above formulae based on the means of the VDW of each treatment or control (replicates = 1000, using the 'boot' function in the 'boot' package [41,42] in R v. 3.4 [43]). Bootstrapped 95% confidence intervals (*Bca* method) were generated for the RIs and net change rates of the VDW of each group. RIs and net change rates of the VDW were considered significantly different between groups when the 95% confidence intervals did not overlap.

Data were checked for normality using the Shapiro–Wilk test. The Bartlett test was applied to check for variance homogeneity. Planned comparisons of least-squares means (independent *t*-test: *t*-value refers to the estimate divided by the s.e.) were performed to detect differences in RIs, change rates of VDW, standardized sperm area in histological sections and HDW between different recovery periods, the two temperature conditions (9°C and 12°C), and initially electroejaculated and control animals. Planned comparisons were performed with STATISTICA 7.0 (StatSoft, Hamburg). To check for no differences in size of crabs among groups, a one-way ANOVA was performed in R.

# 3. Results

Mean CL of males was 111.7 ± 5.6 mm (range = 98–122 mm). All values in the Results section are means ± standard deviations. No significant difference was detected in size among individuals of all experimental groups (one-way ANOVA, $F_{7,\ 31} = 0{,}481$, $p = 0.841$; $n = 39$).

Initial mean dry weight of paired vasa deferentia (VDW) of *L. santolla* without electroejaculation was 35.3 ± 9.0 mg (range = 26.2–43.4 mg, figure 3). After electroejaculation, the mean VDW accounted for 14.8 ± 7.4 mg (range = 6.8–24.3 mg). Electroejaculation decreased the VDW 58.2% compared to the initial mean value. The mean VSI was 0.047 ± 0.007% at the beginning of the experiment.

Within 30 days after electroejaculation, seminal reserves of *L. santolla* were not fully recovered at any of the experimental temperatures (figure 4). The RI at 9°C increased significantly between 15 and 30 days. After 30 days, the RI was significantly larger in seawater at 9°C compared to 12°C. Also, the change rate of VDW in the control after 30 days was significantly larger at 9°C than at 12°C (figure 5).

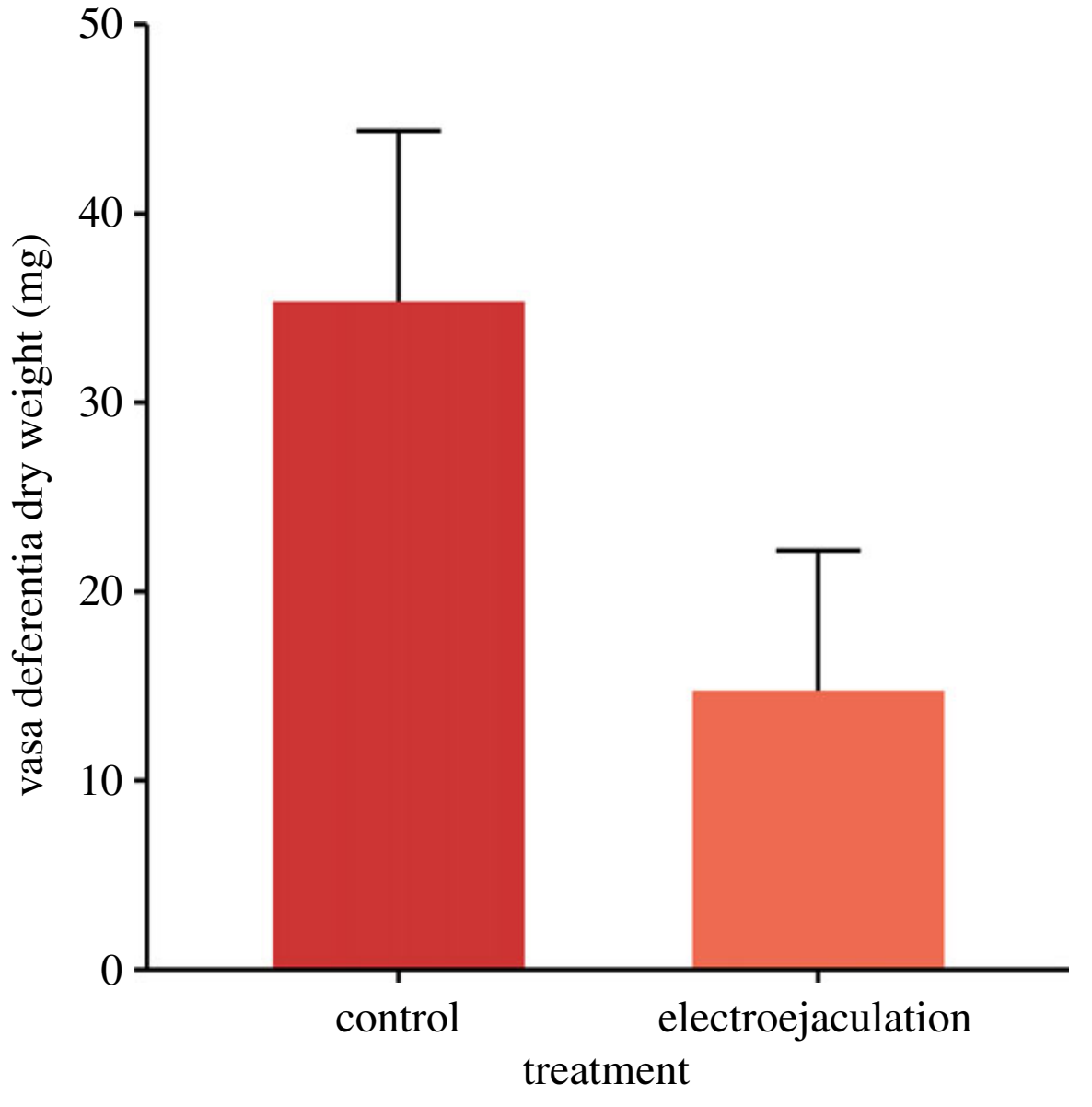


**Figure 3.** Initial vasa deferentia dry weight without and after electroejaculation in *Lithodes santolla* ($n = 4$ and 5 in control and electroejaculated crabs, respectively). Values are means ± standard deviations.

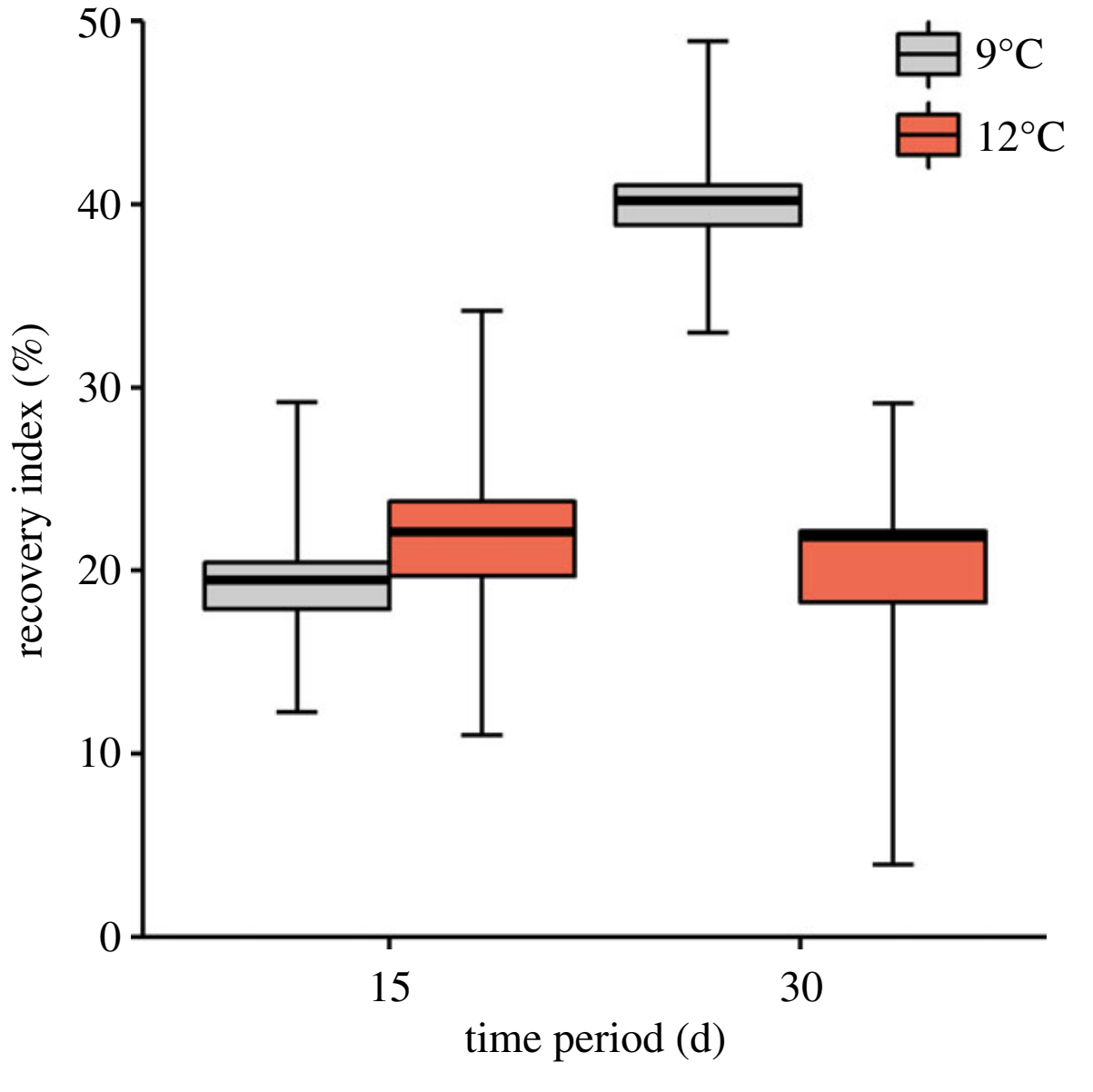


**Figure 4.** Recovery index of the corresponding time periods at 9°C and 12°C in *Lithodes santolla*. Box plot indicates mean, first and third quartiles and 95% confidence interval of the mean (whiskers) estimated from 1000 bootstrap replicates. Calculation of the recovery index was based on $n = 5$ in each group and $n = 4$ in initial group without electroejaculation.

Initial mean standardized sperm area of the paired vasa deferentia was 0.5 ± 0.26 (range = 0.28–0.88; figure 6). Electroejaculation decreased the standardized sperm area significantly compared to the initial value (mean after electroejaculation = 0.2 ± 0.09; range = 0.07–0.26; planned comparison (LSM), $t$-value = 2.72, $p = 0.013$). Standardized sperm area was changed neither within 15 nor 30 days after electroejaculation. Neither after 15 nor 30 days, did a recovery exist in terms of differences in standardized sperm area between 9°C and 12°C.

At the beginning of the experiment, mean dry weight of the hepatopancreas (HDW) and the hepatosomatic index (HSI) were 5.61 ± 2.72 g (range = 2.61–12.24 g) and 7.06 ± 1.77%, respectively. Mean HDW was significantly increased after 15 days only at 12°C (planned comparison (LSM), $t$-value = −2.20, $p = 0.03$; figure 7). After 30 days, mean HDW was significantly increased at both experimental temperatures (9°C: $t$-value = −2.61, $p = 0.01$; and 12°C: $t$-value = −2.62, $p = 0.01$) and hepatopancreas weighed 10.57 ± 1.34 g and 10.02 ± 2.03 g at 9°C and 12°C, respectively. In the control individuals,

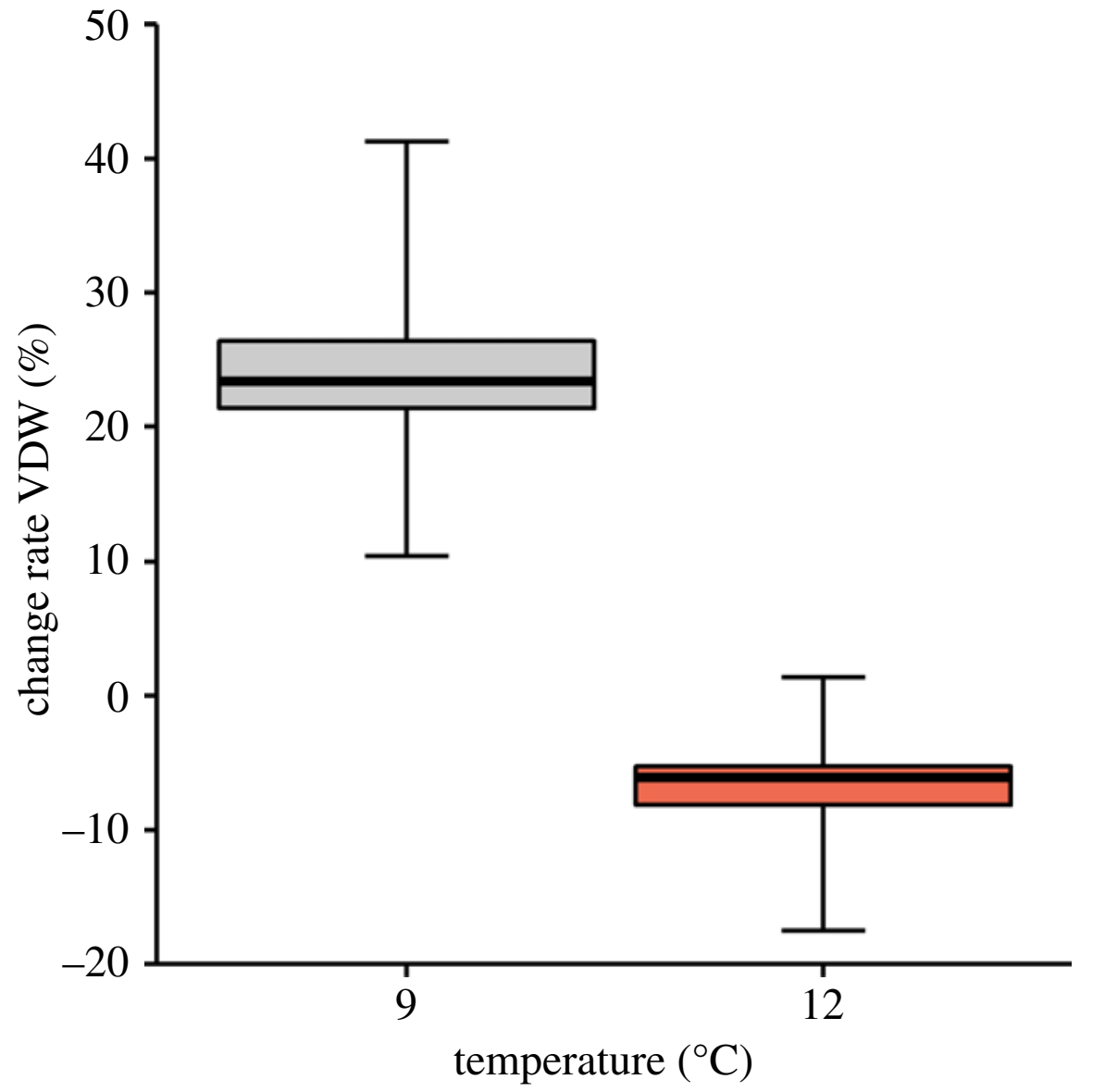


**Figure 5.** Net change rate of vasa deferentia dry weight in control individuals after 30 days. Box plot indicates mean, first and third quartiles and 95% confidence interval of the mean (whiskers) estimated from 1000 bootstrap replicates. Calculation of net change rate of vasa deferentia dry weight was based on $n = 5$ in each group and $n = 4$ in initial group without electroejaculation.

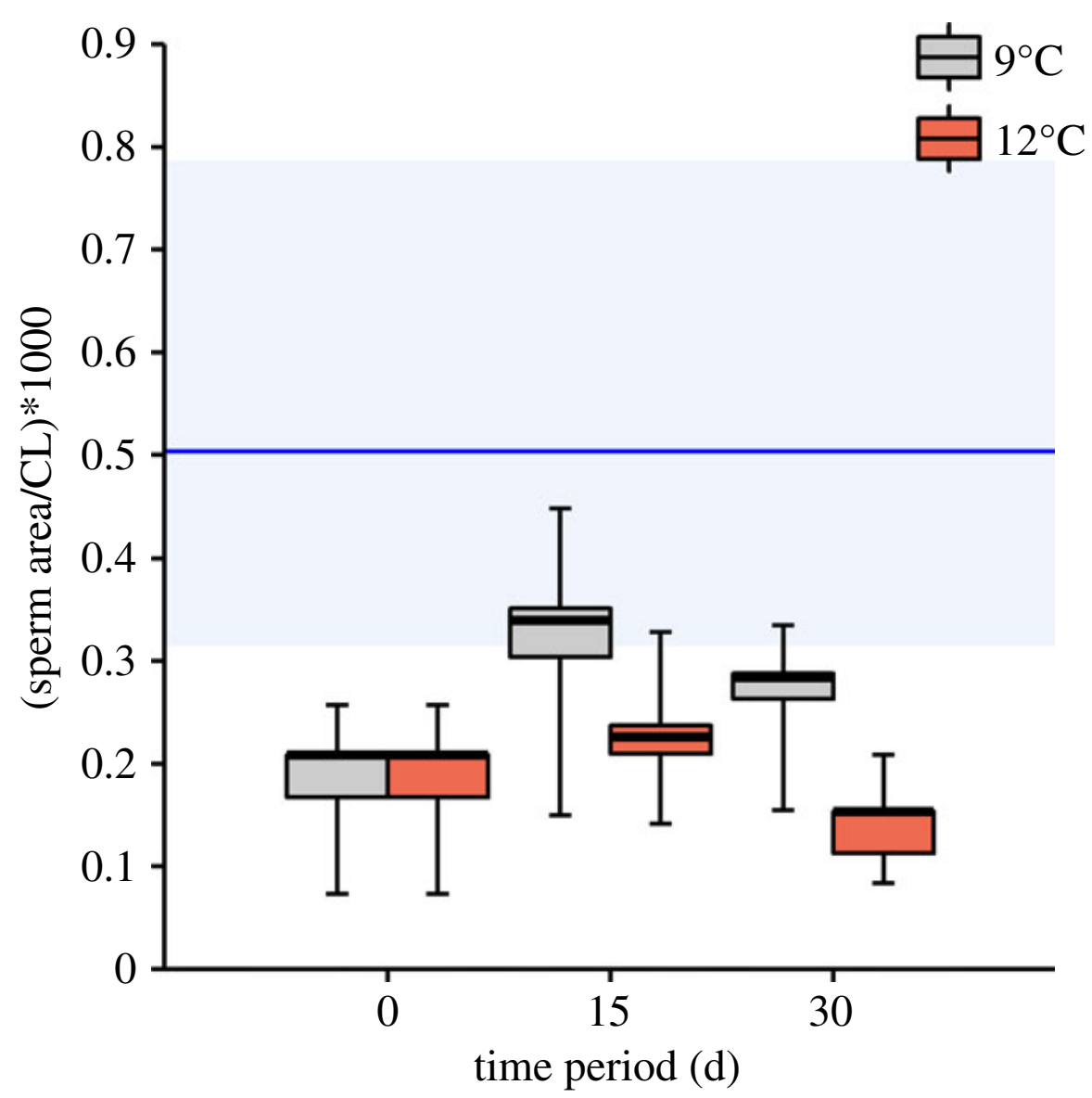


**Figure 6.** Standardized sperm area of paired vasa deferentia of *Lithodes santolla* in initially electroejaculated crabs after the corresponding time periods at 9°C and 12°C ($n = 4$ in each initial group, $n = 5$ in each recovery group, $n = 3$ after 30 days at 12°C). Box plot indicates mean, first and third quartiles and 95% confidence interval of the mean (whiskers). Blue horizontal line represents mean standardized sperm area of initial crabs without electroejaculation ± 95% confidence interval.

HDW was significantly increased after 30 days at 9°C (mean = 11.59 ± 2.00 g; $t$-value = −4.03, $p$ = 0.0002) and 12°C (mean = 12.88 ± 3.33 g; $t$-value = −4.9, $p$ = 0.00002). HDW did not differ after 30 days between initially electroejaculated and control individuals, either at 9°C ($t$-value = −1.42, $p$ = 0.16) or at 12°C ($t$-value = −1.98, $p$ = 0.055).

## 4. Discussion

Sperm and seminal reserves of *Lithodes santolla* were not fully recovered within 30 days after depletion through electroejaculation, either at 9°C or at 12°C. However, temperature significantly affected

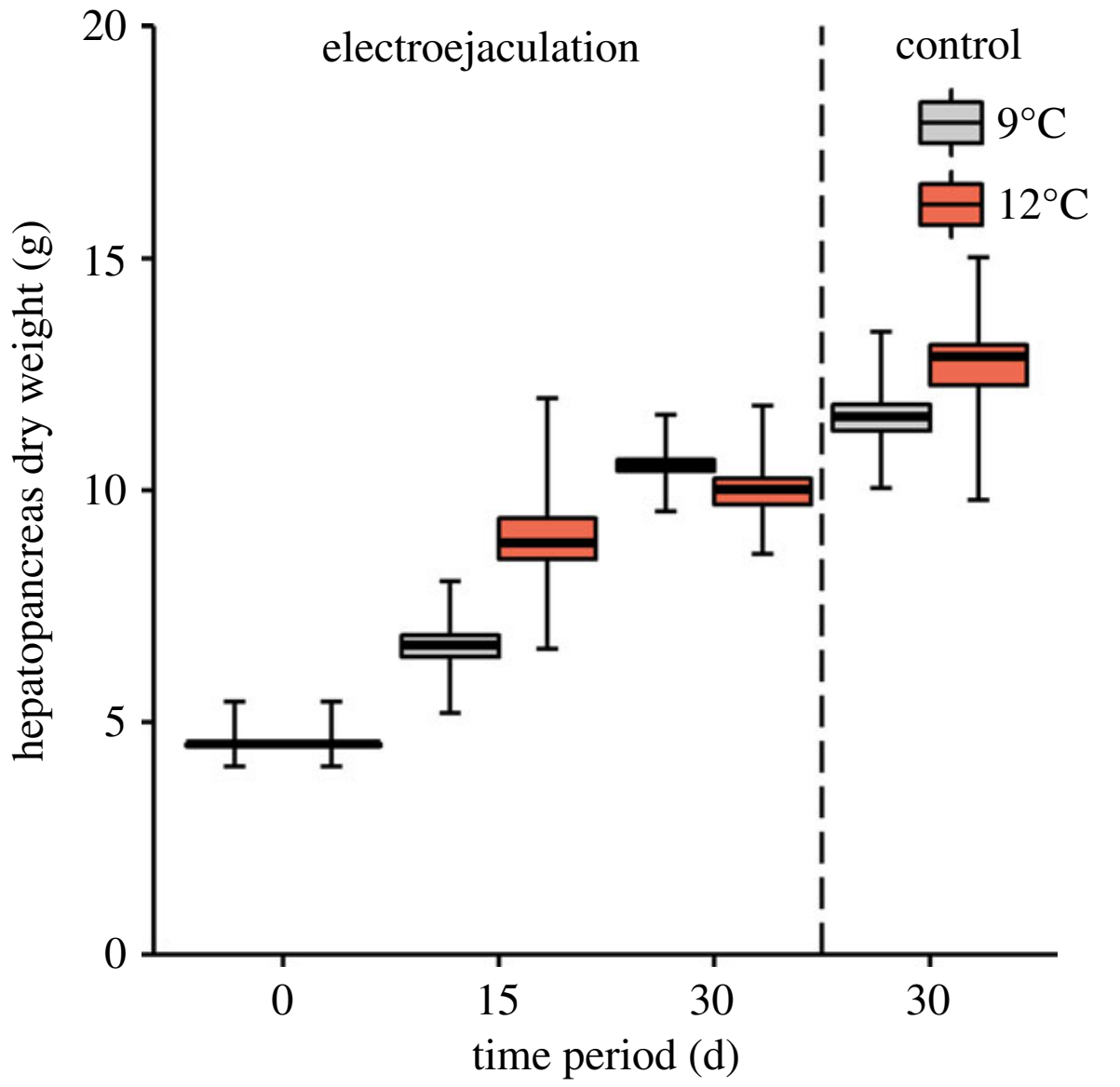


**Figure 7.** Dry weight of the hepatopancreas of *Lithodes santolla* after the corresponding time periods at 9°C and 12°C in initially electroejaculated and control individuals ($n = 5$ in each group; one outlier eliminated in initial group in graphic). Box plot indicates mean, first and third quartiles and 95% confidence interval of the mean (whiskers). Vertical dashed line separates between initially electroejaculated and control individuals.

recovery of seminal stock, and *L. santolla* replenished seminal reserves faster at 9°C than at 12°C. Effect of temperature on sperm recovery was not that pronounced and a trend of increase and decrease of the standardized sperm area at 9°C and 12°C, respectively, was detected.

*Lithodes santolla* can be fished at sites between 5°C and 12°C along its range of distribution. The zone of origin of the individuals in this study is located close to the northern limit of distribution of the species where crabs reproduce in shallow waters and mean temperatures present during mating are 9°C (austral winter) and 12°C (throughout the year). Temperature had a significant effect on the recovery rate of seminal reserves in *L. santolla* after 30 experimental days. The faster seminal material recovery rate in seawater of 9°C than 12°C after 30 days may be explained by the close proximity of the crabs in this study to their northern distributional limit. Hall & Thatje [44] concluded that deep-water representatives of the Lithodinae subfamily were excluded from waters exceeding temperatures of 13°C. The experimental temperature of 12°C may be close to their threshold of temperature tolerance and therefore, crabs may be energetically less efficient. Similarly, 1-year-old juveniles of *L. santolla* can tolerate seawater between 6°C and 15°C; however, their preferred temperature is 9°C [45] with a maximum growth and moulting rate [46,47]. The HDW has doubled during 30 days. At 12°C, crabs started to accumulate energy in the hepatopancreas already after 15 days but then did not immediately invest energy in the reproductive system during the duration of the experiment. Probably at 9°C, energy of the hepatopancreas was converted into seminal material more efficiently.

Temperature-dependent seminal recovery and seasonal variability in the vertical distribution of temperature in the zone of study [48–53] suggest the importance of bathymetrical migrations which could be associated with moving to the optimum temperature to promote recovery of seminal reserves. Seasonal migrations associated with reproduction are a characteristic of lithodid crabs [26] and are well described in lithodid species from the Northern Hemisphere [26]. By contrast, little details are available on these migratory movements in the Southern Hemisphere. Observations of migrations of *L. santolla* exist mainly from the Magellan region and the Atlantic Ocean in Argentina where male and female crabs have been observed to migrate to shallow waters in October and November to reproduce where mating takes place in December and subsequently both sexes return to deeper waters [54]. Timing and magnitude of the pattern of migration in *L. santolla* from northern parts of its distribution is unknown. However, migration of *L. santolla* in the Seno de Reloncaví and interior waters of Chiloé to deeper and cooler waters after reproduction could be substantial to recover seminal material stock in a more efficient form.

Effect of temperature on the recovery of sperm was not that pronounced (non-significant changes) as seminal recovery; however, a trend of increase and decrease in the standardized sperm area in the vasa deferentia lumen was detected at 9°C and 12°C, respectively. We suggest that the estimation of area

covered by sperm in the present study has restrictions due to small sample size and possibly little statistical power.

Our results of incomplete sperm and seminal recovery during 30 days correspond with the previous description of a long spermatogenesis in this species from a different population in the Beagle Channel [27] and in the red king crab *Paralithodes camtschaticus* which inhabits the Northern Hemisphere [55,56]. Direct comparison of the previously described period of spermatogenesis in *L. santolla* and annual pattern of sperm fullness in the vasa deferentia [27] is avoided, due to large latitudinal differences between the study zones and probably varying reproductive traits [33]. Similarly to the results of our recovery rates, in the spiny king crab *P. brevipes* and in the stone crab *Hapalogaster dentata*, sperm numbers in the vasa deferentia are not fully recovered even after 28 and 20 days, respectively, after depletion [16,17]. By contrast, in large males of the brachyuran swimming crab *Callinectes sapidus*, weight of the vasa deferentia is recovered relatively fast and recovery requires 9–20 days after two consecutive matings [7]. The majority of experiments investigating sperm recovery rate in the context of sperm depletion were not conducted under stable experimental temperatures, which makes it difficult to compare results among them (seawater temperature ranges: −0.6–4.3°C [16], 10–19.9°C [17], 0–29°C [7]). By contrast, in aquaculture research, temperature affects the sperm replenishment period in the shrimp *Litopenaeus setiferus* after sperm depletion through electroejaculation (8 days at 25°C, 7 days at 30°C and 6 days at 33°C [19]). In another tropical penaeid, spermatophores are regenerated after manual extrusion in 16 days (*Penaeus brasiliensis*: 27°C [18]).

Apart from fluctuating temperature conditions in some studies, it is difficult to make interspecific comparisons due to inconsistency in the method applied to deplete seminal reserves (electroejaculation versus varying number of successive natural matings) and the type of information reported related to the male reproductive organ (sperm number in the vasa deferentia—vasa deferentia weight—macroscopic visual inspection). The two methods we used to estimate recovery are complementary, as they refer to distinct components of the reproductive system. Dry weight of the vasa deferentia represents the total of sperm and seminal fluids, while the standardized sperm area relates solely to the relative sperm amount. We suggest that the most appropriate method to be applied in further studies depends on the specific research question. In the present study, electroejaculation was a useful, fast and standardized method to artificially induce ejaculation and deplete sperm and seminal material stocks. While our manipulative approach with electroejaculation provides a good basis, it would be favourable in a next step to determine the amount of ejaculate delivered during one mating and the number of possible successive matings for a better understanding of the male reproductive biology and interpretation of our results. Details about lithodid mating ability are described only in *P. camtschaticus* and *P. brevipes* which transfer only portions of their sperm reserves during one mating [4,57]. Lithodid crabs are polygamous. For example, the male king crab *P. camtschaticus* is able to mate with up to seven females and result in full egg clutches [57]. However, the maximum possible mating frequency of males of *L. santolla* is unknown [58]. In *P. brevipes*, an increase in the mating frequency reduces the sperm number ejaculated, the percentage of spawning females and female fertilization rate. After the second successive mating of small males (less than 100 mm CL), partial or null fertilization occurs in females [4]. Crustaceans with slow seminal recovery in a male-only fishery which have mated repeatedly might have depleted seminal reserves especially as the reproductive season progresses. Temporal variation of sperm reserves has been observed in *P. brevipes* and the proportion of depleted males increased throughout the reproductive season [4].

*Lithodes santolla* is distributed over a wide range extending over 19° latitudes in the Pacific [22,23,59]. Along the coast of Chile, the sea surface temperature shows a gradient and is decreasing from north to south [60]. Therefore, *L. santolla* is exposed to regional differences in seawater temperature. The effect of temperature on the seminal recovery rate may generate variations in the reproductive potential of males. Males located close to the northern distributional limit have a slower seminal recovery rate and after successive matings may deplete their reserves faster in contrast to individuals inhabiting more southern or deeper habitats with a temperature closer to 9°C. Considering the predictions of ocean warming over the next decades, this fact could have an even greater impact on the reproductive potential of males in the future [61]. While our results represent a first step towards improving knowledge on male reproductive traits, we highlight the importance of further research on aspects of reproduction of *L. santolla* in its northern distributional limit to allow a sustainable crustacean fishery in the future.

## 5. Conclusion

Summarizing the aspects of the reproductive biology of *L. santolla*, such as absence of a sperm storage organ coupled with slow sperm and seminal recovery, suggest that *L. santolla* populations subject to

intense male-only fisheries may be generally vulnerable to depletion of seminal reserves. Temperature modulated the time necessary to recover seminal reserves. Seminal recovery was faster in seawater of 9°C than that of 12°C, indicating that different seminal replenishment rates in relation to latitude and depth might exist. Especially individuals inhabiting the northern limit of distribution might be susceptible to seminal depletion, in contrast to those located in southern or deeper habitats. Considering that the seminal recovery rate was slower in seawater of 12°C, a climate change scenario could additionally aggravate the risk of seminal depletion in *L. santolla* in its northern distributional limit.

Data accessibility. Data analysed in this study are available online in the electronic supplementary material.

Authors' contributions. K.Pr., L.M.P. and K.Pa. designed the study; K.Pr. and K.Pa. carried out the laboratory work; K.Pr. and L.M.P. analysed the data; K.Pr. wrote the manuscript; L.M.P. and K.Pa. reviewed and revised the manuscript. All authors gave final approval for publication.

Competing interests. We have no competing interests.

Funding. This work was funded by FONDECYT 1150388 and FONDAP 15150003. The author Katrin Pretterebner was funded by a Chilean PhD scholarship from CONICYT.

Acknowledgements. We specially thank Marcela Paz Riveros and Genaro Alvial for their histological sections. We thank Dr Jonas Keiler and an anonymous reviewer for the revision and really appreciate detailed comments which have helped to improve the manuscript.

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
