## [Reviewer comments · Royal Society Open Science]

Review History

RSOS-181700.R0 (Original submission)

Review form: Reviewer 1

Is the manuscript scientifically sound in its present form?

Yes

Are the interpretations and conclusions justified by the results?

Yes

Is the language acceptable?

Yes

Is it clear how to access all supporting data?

Yes

Do you have any ethical concerns with this paper?

No

Have you any concerns about statistical analyses in this paper?

No

Recommendation?

Accept with minor revision (please list in comments)

Comments to the Author(s)

RSOS 181700: Temperature-dependent seminal recovery in the southern king crab
Katrin Pretterebner, Luis Miguel Pardo, Kurt Paschke

General evaluation:

This study well demonstrated the effects of temperature on the recovery rate of spermatophore and seminal reserves in *Lithodes santilla* although data set is somewhat small. Results of this study would help to understand effect of increasing temperature due to climate change (global warming) on reproduction of lithodid crab resources, which would be of broad interest to the readers.

I recommend this MS to be published in Royal Society Open Science.

Minor comments:

[Title]

OK.

[Abstract]

OK.

[Introduction]

L. 80-81: Reproductive season varies depending on the region (between north and southern parts of their distribution)?

[Materials and Methods]

L. 107-109: For readers' understanding, provide picture and/or movie from the time of the electroejaculation.

L. 111-113: Describe time periods when authors conducted first and second experiments

[Results]

OK.

[Discussion]

L. 275-281: As a result, what method do you think the best to estimate the recovery rate of seminal reserves? Please describe.

L. 306-320: Delete the conclusion section because this section (L. 307-320) is redundant.

I hope that my comments will help to improve this manuscript.

Review form: Reviewer 2 (Jonas Keiler)

Is the manuscript scientifically sound in its present form?

Yes

Are the interpretations and conclusions justified by the results?

No

Is the language acceptable?

Yes

Is it clear how to access all supporting data?

Not Applicable

Do you have any ethical concerns with this paper?

No

Have you any concerns about statistical analyses in this paper?

Yes

Recommendation?

Major revision is needed (please make suggestions in comments)

Comments to the Author(s)

Dear editors and authors,

basically the manuscript is interesting and straight forward which makes it generally suitable for publication in RSOS. The present study tries to answer the question if and what way seminal recovery in the southern king crab *Lithodes santolla* is temperature dependent which is of relevance since *L. santolla* is important in fisheries in South America (e.g. in Chile) and only males are caught which leaves fewer males to fertilize females. The authors used a clear experimental setup and compared seminal recovery rates in terms of dry weight differences and differences of histologically assessed sperm area for animals caught and kept at 9°C and 12°C. The manuscript is written in a good English, only some minor things could be rephrased (see attached pdf). Abstract, introduction, m&m and results are written in a clear and comprehensible way. Besides some minor aspects (technical and textual) which should be checked (see attached pdf) and if appropriate rephrased, some questions and contraries have arisen, listed in the following (and in the attached pdf). This concerns strongest the discussion where some conclusions (e.g. increased metabolisms through higher temperature decreases seminal recovery) appeared which I cannot follow without some reliable references and a better argumentation. I suggest to rephrase the major points in the discussion and to check the comments and suggestions in the pdf. To enhance the method with sperm area at least three different regions of each VD should be measured and checked for variability.

With kind regards,

Jonas Keiler

p2: If the authors want to use the plural of sperm I suggest to use the term "sperms"

p4: If the authors really meant "physiologically" they have to explain how they have proved this. But I guess they meant "physically" in terms of body size.

p4: Since the animals are not genetically identical and differ in size and most probably in age it would be more appropriate in most cases to use the standard deviation.

See also e.g. Altman & Bland 2005 BMJ.

p4: What is the usual water temperature during mating season?

p5: How many slices per section were measured for sperm area?

p5: Sperm area should be put in relation to CL and/or diameter of VD.

p7: Why is group sample size 4 while in the method parts the authors wrote 5 per group?

p9: I would say that "brood" has already been fertilized. So better say "eggs".

p9: What about a seasonal aspect in the VDW increase? The authors collected the animals in October and kept them for a month until the experiments started. The mating season normally starts in late Nov to early Dec. So maybe there was either some delay in VD filling due to catching or other artificial factors, or the males of the studied population just usually fill their VDs later. I think a crucial information would be what temperature is normally present in the water during mating!

p9: Is natural ejaculation in *L. santolla* an "all or something" action? So, during copulation, is the whole mature spermatophore packet transferred leaving the VD completely empty? If eggs of several females could be fertilized by one "load" in subsequent copulations what is the disadvantage in this? Or what is the point here?

p11: I am not sure if I understood this correctly: Mating season for *L. santolla* is during southern hemispheric summer (starting late Nov to Dec) when water temperature is normally slightly increased. What is the normal temperature condition during mating in natural environment (and does mating take place at a certain depth?) for *L. santolla*? I do not really understand why higher water temperature of 12°C does effect seminal filling/maturing unfavorable if mating is usually during higher water temperatures (in summer).

At least the author's explanation concerning metabolism is not convincing for me.

p18: N-numbers should be given in all figure legends.

p21: To be honest, this figure is not really informative. At least, areas with sperm should be marked with a transparent color or in a similar way.

Decision letter (RSOS-181700.R0)

09-Nov-2018

Dear Mrs Pretterebner,

The editors assigned to your paper ("Temperature-dependent seminal recovery in the southern king crab") have now received comments from reviewers. We would like you to revise your paper in accordance with the referee and Associate Editor suggestions which can be found below (not including confidential reports to the Editor). Please note this decision does not guarantee eventual acceptance.

Please submit a copy of your revised paper before 02-Dec-2018. Please note that the revision deadline will expire at 00.00am on this date. If we do not hear from you within this time then it will be assumed that the paper has been withdrawn. In exceptional circumstances, extensions may be possible if agreed with the Editorial Office in advance. We do not allow multiple rounds of revision so we urge you to make every effort to fully address all of the comments at this stage. If deemed necessary by the Editors, your manuscript will be sent back to one or more of the original reviewers for assessment. If the original reviewers are not available, we may invite new reviewers.

To revise your manuscript, log into <http://mc.manuscriptcentral.com/rsos> and enter your Author Centre, where you will find your manuscript title listed under "Manuscripts with

Decisions." Under "Actions," click on "Create a Revision." Your manuscript number has been appended to denote a revision. Revise your manuscript and upload a new version through your Author Centre.

- Data accessibility

If you wish to submit your supporting data or code to Dryad (<http://datadryad.org/>), or modify your current submission to dryad, please use the following link:
<http://datadryad.org/submit?journalID=RSOS&manu=RSOS-181700>

- Competing interests

- Authors' contributions

AB carried out the molecular lab work, participated in data analysis, carried out sequence alignments, participated in the design of the study and drafted the manuscript; CD carried out the statistical analyses; EF collected field data; GH conceived of the study, designed the study,

coordinated the study and helped draft the manuscript. All authors gave final approval for publication.

- Acknowledgements

- Funding statement

Please note that Royal Society Open Science charge article processing charges for all new submissions that are accepted for publication. Charges will also apply to papers transferred to Royal Society Open Science from other Royal Society Publishing journals, as well as papers submitted as part of our collaboration with the Royal Society of Chemistry (<http://rsos.royalsocietypublishing.org/chemistry>). If your manuscript is newly submitted and subsequently accepted for publication, you will be asked to pay the article processing charge, unless you request a waiver and this is approved by Royal Society Publishing. You can find out more about the charges at <http://rsos.royalsocietypublishing.org/page/charges>. Should you have any queries, please contact openscience@royalsociety.org.

on behalf of Prof Kevin Padian (Subject Editor)
openscience@royalsociety.org

Editor comments:

Thanks for submitting this manuscript to the journal. The reviewers are broadly favourable towards your work, but each recommends reasonable adjustments prior to further consideration. Good luck in addressing these, and we look forward to receiving your revision shortly.

Comments to Author:

Reviewers' Comments to Author:

Reviewer: 1

Comments to the Author(s)

Royal Society Open Science
RSOS 181700: Temperature-dependent seminal recovery in the southern king crab
Katrin Pretterebner, Luis Miguel Pardo, Kurt Paschke

General evaluation:

This study well demonstrated the effects of temperature on the recovery rate of spermatophore and seminal reserves in *Lithodes santilla* although data set is somewhat small. Results of this study would help to understand effect of increasing temperature due to climate change (global

warming) on reproduction of lithodid crab resources, which would be of broad interest to the readers.

I recommend this MS to be published in Royal Society Open Science.

Minor comments:

[Title]

OK.

[Abstract]

OK.

[Introduction]

L. 80-81: Reproductive season varies depending on the region (between north and southern parts of their distribution)?

[Materials and Methods]

L. 107-109: For readers' understanding, provide picture and/or movie from the time of the electroejaculation.

L. 111-113: Describe time periods when authors conducted first and second experiments

[Results]

OK.

[Discussion]

L. 275-281: As a result, what method do you think the best to estimate the recovery rate of seminal reserves? Please describe.

L. 306-320: Delete the conclusion section because this section (L. 307-320) is redundant.

I hope that my comments will help to improve this manuscript.

Reviewer: 2

Comments to the Author(s)

Dear editors and authors,

basically the manuscript is interesting and straight forward which makes it generally suitable for publication in RSOS. The present study tries to answer the question if and what way seminal recovery in the southern king crab *Lithodes santolla* is temperature dependent which is of relevance since *L. santolla* is important in fisheries in South America (e.g. in Chile) and only males are caught which leaves fewer males to fertilize females. The authors used a clear experimental setup and compared seminal recovery rates in terms of dry weight differences and differences of histologically assessed sperm area for animals caught and kept at 9°C and 12°C. The manuscript is written in a good English, only some minor things could be rephrased (see attached pdf). Abstract, introduction, m&am; and results are written in a clear and comprehensible way. Besides some minor aspects (technical and textual) which should be checked (see attached pdf) and if appropriate rephrased, some questions and contraries have arisen, listed in the following (and in the attached pdf). This concerns strongest the discussion where some conclusions (e.g. increased metabolisms through higher temperature decreases seminal recovery) appeared which I cannot follow without some reliable references and a better argumentation.

I suggest to rephrase the major points in the discussion and to check the comments and suggestions in the pdf. To enhance the method with sperm area at least three different regions of each VD should be measured and checked for variability.

With kind regards,
Jonas Keiler

p2: If the authors want to use the plural of sperm I suggest to use the term "sperms"

p4: If the authors really meant "physiologically" they have to explain how they have proved this. But I guess they meant "physically" in terms of body size.

p4: Since the animals are not genetically identical and differ in size and most probably in age it would be more appropriate in most cases to use the standard deviation.
See also e.g. Altman & Bland 2005 BMJ.

p4: What is the usual water temperature during mating season?

p5: How many slices per section were measured for sperm area?

p5: Sperm area should be put in relation to CL and/or diameter of VD.

p7: Why is group sample size 4 while in the method parts the authors wrote 5 per group?

p9: I would say that "brood" has already been fertilized. So better say "eggs".

p9: What about a seasonal aspect in the VDW increase? The authors collected the animals in October and kept them for a month until the experiments started. The mating season normally starts in late Nov to early Dec. So maybe there was either some delay in VD filling due to catching or other artificial factors, or the males of the studied population just usually fill their VDs later. I think a crucial information would be what temperature is normally present in the water during mating!

p9: Is natural ejaculation in *L. santolla* an "all or something" action? So, during copulation, is the whole mature spermatophore packet transferred leaving the VD completely empty? If eggs of several females could be fertilized by one "load" in subsequent copulations what is the disadvantage in this? Or what is the point here?

p11: I am not sure if I understood this correctly: Mating season for *L. santolla* is during southern hemispheric summer (starting late Nov to Dec) when water temperature is normally slightly increased. What is the normal temperature condition during mating in natural environment (and does mating take place at a certain depth?) for *L. santolla*? I do not really understand why higher water temperature of 12°C does effect seminal filling/maturing unfavorable if mating is usually during higher water temperatures (in summer).

At least the author's explanation concerning metabolism is not convincing for me.

p18: N-numbers should be given in all figure legends.

p21: To be honest, this figure is not really informative. At least, areas with sperm should be marked with a transparent color or in a similar way.

Author's Response to Decision Letter for (RSOS-181700.R0)

See Appendix A.

RSOS-181700.R1 (Revision)

Review form: Reviewer 2 (Jonas Keiler)

Is the manuscript scientifically sound in its present form?

Yes

Are the interpretations and conclusions justified by the results?

Yes

Is the language acceptable?

Yes

Is it clear how to access all supporting data?

Not Applicable

Do you have any ethical concerns with this paper?

No

Have you any concerns about statistical analyses in this paper?

No

Recommendation?

Accept with minor revision (please list in comments)

Comments to the Author(s)

Dear editors and authors,

The revised version of the manuscript was basically improved following the reviewer's comments and suggestions.

Although I am still not convinced of the hypothesis that a decreased metabolism increases seminal production by providing more energy for this process, the actual version of the text provides interesting data on seminal depletion in the commercially import king crab *L. santolla* and potentially useful aspects for more sustainable fisheries. However, the latter point could be provided with a clearer statement on how or if at all the author's findings suggest improvements/changes for fishery of *L. santolla*.

I suggest the present manuscript to be published in RSOS after correcting the following points and further remarks in the pdf attached (Appendix B).

With kind regards,

Jonas Keiler

p9 l 227: The addition of the annual mean is confusing here since mating takes place at 9°C during winter, only. The annual mean does not play a role during mating, does it?

figure page 17: The gonopores are at the coxae!

Decision letter (RSOS-181700.R1)

17-Dec-2018

Dear Mrs Pretterebner:

On behalf of the Editors, I am pleased to inform you that your Manuscript RSOS-181700.R1 entitled "Temperature-dependent seminal recovery in the southern king crab *Lithodes santolla*" has been accepted for publication in Royal Society Open Science subject to minor revision in accordance with the referee suggestions. Please find the referees' comments at the end of this email.

The reviewers and Subject Editor have recommended publication, but also suggest some minor revisions to your manuscript. Therefore, I invite you to respond to the comments and revise your manuscript.

- Ethics statement

- Data accessibility

If you wish to submit your supporting data or code to Dryad (<http://datadryad.org/>), or modify your current submission to dryad, please use the following link:
<http://datadryad.org/submit?journalID=RSOS&manu=RSOS-181700.R1>

- Competing interests

- Authors' contributions

- Acknowledgements

- Funding statement

Because the schedule for publication is very tight, it is a condition of publication that you submit the revised version of your manuscript before 26-Dec-2018. Please note that the revision deadline will expire at 00.00am on this date. If you do not think you will be able to meet this date please let me know immediately.

Supplementary files will be published alongside the paper on the journal website and posted on

the online figshare repository (<https://figshare.com>). The heading and legend provided for each supplementary file during the submission process will be used to create the figshare page, so please ensure these are accurate and informative so that your files can be found in searches. Files on figshare will be made available approximately one week before the accompanying article so that the supplementary material can be attributed a unique DOI.

Please note that Royal Society Open Science charge article processing charges for all new submissions that are accepted for publication. Charges will also apply to papers transferred to Royal Society Open Science from other Royal Society Publishing journals, as well as papers submitted as part of our collaboration with the Royal Society of Chemistry (<http://rsos.royalsocietypublishing.org/chemistry>). If your manuscript is newly submitted and subsequently accepted for publication, you will be asked to pay the article processing charge, unless you request a waiver and this is approved by Royal Society Publishing. You can find out more about the charges at <http://rsos.royalsocietypublishing.org/page/charges>. Should you have any queries, please contact openscience@royalsociety.org.

on behalf of Prof Kevin Padian (Subject Editor)
openscience@royalsociety.org

Associate Editor Comments to Author:

The reviewer recommends the paper be accepted once you have addressed their remaining concerns. Please ensure that in your revision you tackle these concerns, and provide a point-by-point response detailing how you have done so. We look forward to receiving your resubmission.

Reviewer comments to Author:

Reviewer: 2

Comments to the Author(s)

Dear editors and authors,
the revised version of the manuscript was basically improved following the reviewer's comments and suggestions.

Although I am still not convinced of the hypothesis that a decreased metabolism increases seminal production by providing more energy for this process, the actual version of the text provides interesting data on seminal depletion in the commercially import king crab *L. santolla* and potentially useful aspects for more sustainable fisheries. However, the latter point could be provided with a clearer statement on how or if at all the author's findings suggest improvements/changes for fishery of *L. santolla*.

I suggest the present manuscript to be published in RSOS after correcting the following points and further remarks in the pdf attached.

With kind regards,

Jonas Keiler

p9 l 227: The addition of the annual mean is confusing here since mating takes place at 9°C during winter, only. The annual mean does not play a role during mating, does it?

figure page 17: The gonopores are at the coxae!

Author's Response to Decision Letter for (RSOS-181700.R1)

See Appendix C.

Decision letter (RSOS-181700.R2)

08-Jan-2019

Dear Mrs Pretterebner,

I am pleased to inform you that your manuscript entitled "Temperature-dependent seminal recovery in the southern king crab *Lithodes santolla*" is now accepted for publication in Royal Society Open Science.

on behalf of Professor Kevin Padian (Subject Editor)
openscience@royalsociety.org

Appendix A

Responses to Reviewer Comments

RSOS 181700: Temperature-dependent seminal recovery in the southern king crab *Lithodes santolla*

Katrin Pretterebner, Luis Miguel Pardo, Kurt Paschke

We would like to thank the reviewers for the revision and really appreciate their detailed comments to improve the manuscript. We have responded to all of the reviewers' comments by making appropriate modifications to our manuscript.

Responses to Reviewer 1

L. 80-81: Reproductive season varies depending on the region (between north and southern parts of their distribution)?

We included the following phrases in the introduction for better understanding: “Biological information of *L. santolla* is largely restricted to the southern limit of distribution (Beagle Channel) without data describing reproductive aspects of males derived from their northern distributional limit. Intraspecific latitudinal variability in the reproductive pattern and traits has been documented in crustacean species. Consequently, *L. santolla* in its northern distributional limit (this study) is exposed to different prevailing environmental conditions (e.g. temperature, photoperiod) which possibly have led to variations in the reproductive pattern between the southern and northern limits of its distribution.”.

[Materials and Methods]

L. 107-109: For readers' understanding, provide picture and/or movie from the time of the electroejaculation.

Figure 1a shows a picture of the placement of electrodes during electroejaculation and gonopores where ejaculate is extruded. For further details, methods are described in Soundarapandian (2013) which is cited.

L. 111-113: Describe time periods when authors conducted first and second experiments

The following phrase was added to Material and Methods: "Both temperature conditions were conducted simultaneously from mid-November to mid-December 2016."

[Results]

OK.

[Discussion]

L. 275-281: As a result, what method do you think the best to estimate the recovery rate of seminal reserves? Please describe.

We included that following section in the discussion: "We used the two methods to estimate recovery complementary, as they refer to distinct components of the reproductive structures. Dry weight of the vasa deferentia represents the total of sperm and seminal fluids, while the standardized sperm area relates solely to the relative sperm amount. We suggest that the most appropriate method to be applied in further studies depends on the specific research question."

L. 306-320: Delete the conclusion section because this section (L. 307-320) is redundant.

The conclusion section was shorted and now is more concise summarizing only the most important findings of the study. The conclusion section was not deleted completely to coincide with the format of the journal Royal Society Open Science.

Responses to Reviewer 2

p2: If the authors want to use the plural of sperm I suggest to use the term "sperms".

Now, the term "sperm" is used in singular. It was corrected in all relevant phrases.

p4: If the authors really meant "physiologically" they have to explain how they have proved this. But I guess they meant "physically" in terms of body size. The right term is "physically".

Indeed we wanted to refer to "physiologically" mature and included the corresponding citation. "Physiologically" mature refers to individuals which are able to develop gonads. Southern king crab males reach physiological maturity in the Beagle channel at a size of 75 mm CL. The crabs used in the experiment were ≥ 98 mm CL and therefore can be considered as "physiologically" mature.

p4: Since the animals are not genetically identical and differ in size and most probably in age it would be more appropriate in most cases to use the standard

deviation.

See also e.g. Altman & Bland 2005 BMJ.

As suggested by the reviewer, instead of using the standard error, we now state the mean with the standard deviation in the results section and in figure 2.

p4: What is the usual water temperature during mating season?

We indicated the usual water temperature during the mating season and included the following phrase: "The zone of origin of the individuals of this study is located close to its northern limit of distribution where crabs reproduce in shallow waters and mean temperatures present during mating are 9°C (winter mean) and 12°C (annual mean).".

p5: How many slices per section were measured for sperm area?

We added the following phrase to Material and Methods: "One slice of the middle section of the vasa deferentia, which corresponds to the section where spermatophore reserves are located, was used to estimate area covered by sperm in the lumen".

p5: Sperm area should be put in relation to CL and/or diameter of VD.

As suggested by the reviewer, now we present [(sperm area/CL) *1000] to avoid influence of crab size.

p7: Why is group sample size 4 while in the method parts the authors wrote 5 per group?

Sample sizes are now stated explicitly for each group.

p9: I would say that "brood" has already been fertilized. So better say "eggs".

We thank the reviewer for this observation. “Brood” was changed to “oocytes”.

p9: What about a seasonal aspect in the VDW increase? The authors collected the animals in October and kept them for a month until the experiments started. The mating season normally starts in late Nov to early Dec. So maybe there was either some delay in VD filling due to catching or other artificial factors, or the males of the studied population just usually fill their VDs later. I think a crucial information would be what temperature is normally present in the water during mating!

In the Beagle Channel the mating season of *L. santolla* starts in late November to early December and lasts approximately one month. However, biological information of *L. santolla* is largely restricted to the southern limit of distribution (Beagle Channel) without data describing reproductive aspects of males derived from their northern distributional limit. Intraspecific latitudinal variability in the reproductive pattern and traits has been documented in crustacean species. Consequently, *L. santolla* in its northern distributional limit (this study) is exposed to different prevailing environmental conditions (e.g. temperature, photoperiod) which possibly have led to variations in the reproductive pattern between the southern and northern limits of its distribution, and *L. santolla* in its northern distributional limit possibly has a different window of reproduction. For this reason we did not conclude a seasonal aspect of seminal recovery in *L. santolla* in its northern limit of distribution. As requested we included the usual water temperature during the mating season assuming a larger window of reproduction of this species but from the northern limit of its distribution and included the following phrase: “The zone of origin of the individuals of this study is located close to its northern limit of distribution where crabs reproduce in

shallow waters and mean temperatures present during mating are 9°C (winter mean) and 12°C (annual mean).”.

p9: Is natural ejaculation in *L. santolla* an "all or something" action? So, during copulation, is the whole mature spermatophore packet transferred leaving the VD completely empty? If eggs of several females could be fertilized by one "load" in subsequent copulations what is the disadvantage in this? Or what is the point here?

In general, we restructured the discussion and where necessary rephrased for more clarity. In *L. santolla* the amount of ejaculate delivered during one mating is unknown. Details about lithodid mating ability are described only in *Paralithodes camtschaticus* and *P. brevipes* which partially transfer sperm of their reserves during one mating resembling a “something” action. Lithodid crabs are polygamous. The crucial point is that, for example, in *P. brevipes* an increase in the mating frequency reduces the sperm number ejaculated, the percentage of spawning females and female fertilization rate. After the second successive mating of small males (<100 mm CL) partial or null fertilization occurs in females. Consequently, crustaceans with slow seminal recovery in a male-only fishery which have mated repeatedly might have depleted seminal reserves especially as the reproductive season progresses.

p11: I am not sure if I understood this correctly: Mating season for *L. santolla* is during southern hemispheric summer (starting late Nov to Dec) when water temperature is normally slightly increased. What is the normal temperature condition during mating in natural environment (and does mating take place at a certain depth?) for *L. santolla*? I do not really understand why higher water temperature of 12°C does effect seminal filling/maturing unfavorable if mating is usually during higher water temperatures (in summer).

At least the author's explanation concerning metabolism is not convincing for me.

Lithodid crabs perform bathymetrical migrations which are associated with reproduction, where mating takes place in shallow waters. These reproductive migrations have been observed in the Beagle channel but information is fragmented. The zone of origin of the individuals of this study is located close to its northern limit of distribution where crabs reproduce in shallow waters and mean temperatures present during mating are 9°C (winter mean) and 12°C (annual mean) (see explanation above for likely varying reproductive pattern between the southern and northern limit of distribution).

L. santolla can be fished at sites between 5°C and 12°C along its range of distribution. The higher water temperature of 12°C likely affects recovery seminal reserves unfavorable because the origin of the crabs from our study is from its northern distributional limit and 12°C may be close to the threshold of temperature tolerance of *L. santolla*. For example, deep-water representatives of the Lithodinae sub-family were excluded from waters exceeding temperatures of 13°C.

We added two reliable references (book section of Willmer et al. 2005 and Paper of Pörtner 2002) to justify that lower temperature implies a reduction of the standard metabolic rate (SMR) of the organism and to better explain that therefore more energy may be available to be assigned to reproduction (faster seminal recovery at 9°C).

p18: N-numbers should be given in all figure legends.

We added N-numbers to all figure legends.

p21: To be honest, this figure is not really informative. At least, areas with sperm should be marked with a transparent color or in a similar way.

We modified figure 5 for better visibility of sperm areas. Now, figure 5 now shows an overview of a histological cross-section of the vasa deferentia of *L. santolla* and a detailed view thereof displaying spermatophores filled with sperm.

Appendix B**ROYAL SOCIETY
OPEN SCIENCE****Temperature-dependent seminal recovery in the southern
king crab *Lithodes santolla***

Journal:	Royal Society Open Science
Manuscript ID	RSOS-181700.R1
Article Type:	Research
Date Submitted by the Author:	29-Nov-2018
Complete List of Authors:	Pretterebner, Katrin; Universidad Austral de Chile Programa de Doctorado en Biología Marina, Facultad de Ciencias; Universidad Austral de Chile Instituto de Ciencias Marinas y Limnológicas, Facultad de Ciencias; Centro de Investigación de Dinámica de Ecosistemas Marinos de Altas Latitudes (IDEAL) Pardo, Luis Miguel; Universidad Austral de Chile Instituto de Ciencias Marinas y Limnológicas, Facultad de Ciencias; 
[revised manuscript text omitted]

$\text{Change rate VDW} (\%) = \frac{\text{VDW}_{C_{t_1}} - \text{VDW}_{C_{t_0}}}{\text{VDW}_{C_{t_0}}} * 100$

Both, RI and net change rates of VDW in control individuals, were estimated through bootstrapping of the
 formulas above based on the means of the VDW of each treatment or control (replicates = 1000, using the ‘boot’
 function in the ‘boot’ package [41,42] in R version 3.4 [43]). Bootstrapped 95% confidence intervals (*Bca* method)
 were generated for the RIs and net change rates of the VDW of each treatment. RIs and net change rates of the
 VDW were considered significantly different between groups when the 95% confidence intervals did not overlap.
 Data were checked for normality using the Shapiro-Wilk test. The Bartlett test was applied to check for variance
 homogeneity. Planned comparisons of least squares means (independent *t*-test: *t*-value refers to the estimate divided
 by the s.e.) were performed to detect differences in RIs, change rates of VDW, standardized sperm area in
 histological sections and HDW between different recovery periods, the two temperature conditions (9°C and 12°C),
 and initially electroejaculated and control animals. Planned comparisons were performed with STATISTICA 7.0
 (StatSoft, Hamburg). To check for no differences in size of crabs among treatments a one-way ANOVA was
 performed in R.

3. Results

Mean CL of males was 111.7 ± 5.6 mm (range = 98–122 mm). All values in the results section are means \pm
 standard deviations. No significant difference was detected in size among individuals of all experimental groups
 (one-way ANOVA, $F_{7, 31} = 0.481$, $p = 0.841$; $n = 39$).

Seminal recovery in king crab

8

Initial mean dry weight of paired vasa deferentia (VDW) of *L. santolla* without electroejaculation was 35.3
± 9.0 mg (range = 26.2–43.4 mg, figure 3). After electroejaculation the mean VDW accounted for 14.8 ± 7.4 mg
(range = 6.8–24.3 mg). Electroejaculation decreased the VDW 58.2% compared to the initial mean value. The mean
vasosomatic index (VSI) at the beginning of the experiment was $0.047 \pm 0.007\%$.

Seminal reserves of *L. santolla* were ~~not fully recovered within 30 days after electroejaculation~~ at none of
the experimental temperatures (figure 4). The recovery index (RI) at 9°C increased significantly between 15 and 30
202 days. After 30 days the RI was significantly larger in seawater of 9°C compared to 12°C. Also, the change rate of
203 VDW in the control after 30 days was significantly larger at 9°C than 12°C (figure 5).

Initial mean standardized sperm area of the paired vasa deferentia was 0.5 ± 0.26 (range = 0.28–0.88; figure
6). Electroejaculation decreased the standardized sperm area significantly compared to the initial value (mean after
electroejaculation = 0.2 ± 0.09 ; range = 0.07–0.26; planned comparison (LSM), t -value = 2.72, $p = 0.013$).
Standardized sperm area was changed neither within 15 nor 30 days after electroejaculation. Neither after 15 nor 30
208 days a recovery existed in terms of differences in standardized sperm area between 9°C and 12°C.

209 At the beginning of the experiment mean dry weight of the hepatopancreas (HDW) and the hepatosomatic
index (HSI) were 5.61 ± 2.72 g (range = 2.61–12.24 g) and $7.06 \pm 1.77\%$, respectively. Mean HDW was
significantly increased after 15 days only at 12°C (planned comparison (LSM), t -value = -2.20, $p = 0.03$; figure 7).
After 30 days mean HDW was significantly increased at both experimental temperatures (9°C: t -value = -2.61, $p =$
0.01; and 12°C: t -value = -2.62, $p = 0.01$) and hepatopancreas weighed 10.57 ± 1.34 g and 10.02 ± 2.03 g at 9°C and
12°C, respectively. In the control individuals HDW was significantly increased after 30 days at 9°C (mean = $11.59 \pm$
2.00 g; t -value = -4.03, $p = 0.0002$) and 12°C (mean = 12.88 ± 3.33 g; t -value = -4.9, $p = 0.00002$). HDW did not
differ after 30 days between initially electroejaculated and control individuals, neither at 9°C (t -value = -1.42, $p =$
0.16) nor 12°C (t -value = -1.98, $p = 0.055$).

4. Discussion

Sperm and seminal reserves of *L. santolla* were not fully recovered within 30 days after depletion through electroejaculation, neither at 9°C nor 12°C. However, temperature significantly affected recovery of seminal stock and *L. santolla* replenished seminal reserves faster at 9°C than 12°C. Effect of temperature on sperm recovery was not that pronounced and a trend of increase and decrease of the standardized sperm area at 9°C and 12°C, respectively, was detected.

L. santolla can be fished at sites between 5°C and 12°C along its range of distribution. The zone of origin of the individuals of this study is located close to its northern limit of distribution where crabs reproduce in shallow waters and mean temperatures present during mating are 9°C (winter mean) and 12°C (annual mean). Hall and Thatje (2009) concluded that deep-water representatives of the Lithodinae sub-family were excluded from waters exceeding temperatures of 13°C. The experimental temperature 12°C may be close to the threshold of temperature tolerance of *L. santolla*. Temperature had a significant effect on the recovery rate of seminal reserves in *L. santolla* after 30 experimental days. The faster seminal material recovery rate in seawater of 9°C than 12°C after 30 days may be explained by the consequences of a higher metabolic rate at 12°C. It is known that, lower temperature implies a reduction of the standard metabolic rate (SMR) of the organism [45,46] and in this condition likely more energy is available to be assigned to reproduction. The HDW has nearly 
[revised manuscript text omitted]

fact could impact even greater the reproductive potential of males if the predictions of ocean warming for the next
decades ~~were~~ correct [63].

5. Conclusion

Summarizing the aspects of reproductive biology of *L. santolla*, such as absence of sperm storage organ coupled with slow sperm- and seminal recovery suggest that *L. santolla* populations subject to intense male-only fisheries may be generally vulnerable to depletion of seminal reserves. Temperature modulates the time necessary to recover seminal reserves. Seminal recovery is faster in seawater of 9°C than 12°C, indicating that different seminal replenishment rates in relation to latitude and depth might exist. Especially individuals inhabiting the northern limit of distribution might be susceptible to seminal depletion, in contrast to those located in southern or deeper habitats. Considering that the seminal recovery rate is slower in seawater of 12°C, a climate change scenario could additionally aggravate the risk of seminal depletion in *L. santolla* in its northern distributional limit.

Animal Ethics

No ethical statement was required prior to conducting research.

Permission to carry out fieldwork

No permissions were required prior to conducting research.

Data Accessibility

Data analysed in this study is available online in the electronic supplementary material.

Authors' Contributions

K Pretterebner, LM Pardo and K Paschke designed the study; K Pretterebner and K Paschke carried out the lab work; K Pretterebner and LM Pardo analyzed the data; K Pretterebner wrote the manuscript; LM Pardo and K Paschke reviewed and revised the manuscript. All authors gave final approval for publication.

Competing Interests

We have no competing interests.

Funding

This work was funded by FONDECYT 1150388 and FONDAP 15150003. The author Katrin Pretterebner was funded by a Chilean PhD scholarship from CONICYT.

Acknowledgements

We especially thank Marcela Paz Riveros and Genaro Alvial for histological sections. We thank Dr. Jonas Keiler and an anonymous reviewer for the revision and really appreciate detailed comments which have helped to improve the manuscript.

References

- 1. Orensanz JM, Armstrong J, Armstrong D, Hilborn R. 1998 Crustacean resources are vulnerable to serial
depletion - The multifaceted decline of crab and shrimp fisheries in the Greater Gulf of Alaska. *Rev. Fish*
*Biol. Fish.* **8**, 117–176. (doi:10.1023/A:1008891412756)
- 2. Hines AH, Jivoff PR, Bushmann PJ, Van Montfrans J, Reed SA, Wolcott DL, Wolcott TG. 2003 Evidence
for sperm limitation in the blue crab, *Callinectes sapidus*. *Bull. Mar. Sci.* **72**, 287–310.
- 3. Pardo LM, Riveros MP, Fuentes JP, Pinochet R, Cárdenas C, Sainte-Marie B. 2017 High fishing intensity
reduces females' sperm reserve and brood fecundity in a eubrachyuran crab subject to sex-and size-biased
harvest. *ICES J. Mar. Sci.* **74**, 2459–2469. (doi:10.1093/icesjms/fsx077)
- 4. Sato T, Ashidate M, Wada S, Goshima S. 2005 Effects of male mating frequency and male size on ejaculate
size and reproductive success of female spiny king crab *Paralithodes brevipes*. *Mar. Ecol. Prog. Ser.* **296**,
251–262. (doi:10.3354/meps296251)
- 5. Hankin DG, Butler TH, Wold PW, Xue QL. 1997 Does intense fishing on males impair mating success of
female Dungeness crab? *Can. J. Fish. Aquat. Sci.* **54**, 655–669. (doi:10.1139/f96-308)
- 6. Sainte-Marie B. 1993 Reproductive Cycle and Fecundity of Primiparous and Multiparous Female Snow
Crab, *Chionoecetes opilio*, in the Northwest Gulf of Saint Lawrence. *Can. J. Fish. Aquat. Sci.* **50**, 2147–
2156. (doi:10.1139/f93-240)
- 7. Kendall MS, Wolcott DL, Wolcott TG, Hines AH. 2001 Reproductive potential of individual male blue
crabs, *Callinectes sapidus*, in a fished population: depletion and recovery of sperm number and seminal
fluid. *Can. J. Fish. Aquat. Sci.* **58**, 1168–1177. (doi:10.1139/cjfas-58-6-1168)
- 8. Pardo LM, Rosas Y, Fuentes JP, Riveros MP, Chaparro OR. 2015 Fishery induces sperm depletion and
reduction in male reproductive potential for crab species under male-biased harvest strategy. *PLoS One* **10**,
1–16. (doi:10.1371/journal.pone.0115525)
- 9. Rondeau A, Sainte-Maire B. 2001 Variable Mate-Guarding Time and Sperm Allocation by Male Snow
Crabs (*Chionoecetes opilio*) in Response to Sexual Competition, and their Impact on the Mating Success of
Females. *Biol. Bull.* **201**, 204–217. (doi:10.2307/1543335)
- 10. Sato T. 2011 Plausible causes for sperm-store variations in the coconut crab *Birgus latro* under large male-
selective harvesting. *Aquat. Biol.* **13**, 11–19. (doi:10.3354/ab00350)
- 11. Sato T, Ashidate M, Jinbo T, Goshima S. 2007 Does male-only fishing influence reproductive success of the
female spiny king crab, *Paralithodes brevipes*? *Can. J. Fish. Aquat. Sci.* **64**, 735–742. (doi:10.1139/f07-
044)
- 12. Parker GA. 1984 Sperm competition and the evolution of animal strategies. In *Sperm Competition and the*
*Evolution of Animal Mating Systems* (ed RL Smith), pp. 1–60. Orlando: Academic Press.
(doi:10.1016/B978-0-12-652570-0.50008-7)
- 13. Levitan DR, Petersen C. 1995 Sperm limitation in the sea. *Trends Ecol. Evol.* **10**, 228–231.
(doi:10.1016/S0169-5347(00)89071-0)
- 14. Dewsbury DA. 1982 Ejaculate cost and male choice. *Am. Nat.* **119**, 601–610. (doi:10.1086/283938)
- 15. Sainte-Marie B. 2007 Sperm Demand and Allocation in Decapod Crustaceans. In *Evolutionary Ecology of*
*Social and Sexual Systems: Crustaceans as Model Organisms* (eds JE Duffy, M Thiel), pp. 191–210. New
York: Oxford University Press. (doi:10.1093/acprof:oso/9780195179927.003.0009)
- 16. Sato T, Ashidate M, Jinbo T, Goshima S. 2006 Variation of sperm allocation with male size and recovery
rate of sperm numbers in spiny king crab *Paralithodes brevipes*. *Mar. Ecol. Prog. Ser.* **312**, 189–199.
(doi:10.3354/meps312189)
- 17. Sato T, Goshima S. 2006 Impacts of male-only fishing and sperm limitation in manipulated populations of
an unfished crab, *Hapalogaster dentata*. *Mar. Ecol. Prog. Ser.* **313**, 193–204. (doi:10.3354/meps313193)
- 18. Braga A, Lopes DLA, Poersch LH, Wasielesky W. 2014 Spermatophore replacement of pink shrimp
*Farfantepenaeus brasiliensis* after manual extrusion: Effect of molting. *Aquaculture* **433**, 313–317.
(doi:10.1016/j.aquaculture.2014.06.032)
- 19. Pascual C, Valera E, Re-Regis C, Gaxiola G, Sanchez A, Ramos L, Soto LA, Rosas C. 1998 Effect of Water
Temperature on Reproductive Tract Condition of *Penaeus setiferus* Adult Males. *J. World Aquac.* **29**, 477–
484. (doi:10.1111/j.1749-7345.1998.tb00672.x)
- 20. Bugnot AB, López Greco LS. 2009 Sperm production in the red claw crayfish *Cherax quadricarinatus*

[revised manuscript text omitted]

Initial vasa deferentia dry weight without and after electroejaculation, respectively, in *Lithodes santolla* ($N = 4-5$ each). Values are means \pm standard deviations.

83x83mm (300 x 300 DPI)

Recovery index of the corresponding time periods at 9°C and 12°C in *Lithodes santolla* ($N = 4-5$ each). Box plot indicates mean, first and third quartiles and 95% confidence interval of the mean (whiskers).

83x83mm (300 x 300 DPI)

Net change rate of vasa deferentia dry weight in control individuals after 30 days ($N = 5$ each). Box plot indicates mean, first and third quartiles and 95% confidence interval of the mean (whiskers).

83x83mm (300 x 300 DPI)

Standardized sperm area of paired vasa deferentia of *Lithodes santolla* in initially electroejaculated crabs after the corresponding time periods at 9°C and 12°C ($N = 3-5$ each). Box plot indicates mean, first and third quartiles and 95% confidence interval of the mean (whiskers). Blue horizontal line represents mean standardized sperm area of initial crabs without electroejaculation \pm 95% confidence interval.

83x83mm (300 x 300 DPI)

Dry weight of the hepatopancreas of *Lithodes santolla* after the corresponding time periods at 9°C and 12°C in initially electroejaculated and control individuals ($N = 5$ each; one outlier eliminated in initial group in graphic). Box plot indicates mean, first and third quartiles and 95% confidence interval of the mean (whiskers). Vertical dashed line separates between initially electroejaculated and control individuals.

83x83mm (300 x 300 DPI)

Responses to Reviewer Comments

RSOS 181700: Temperature-dependent seminal recovery in the southern king crab *Lithodes santolla*

Katrin Pretterebner, Luis Miguel Pardo, Kurt Paschke

We would like to thank the reviewers for the revision and really appreciate their detailed comments to improve the manuscript. We have responded to all of the reviewers' comments by making appropriate modifications to our manuscript.

Responses to Reviewer 1

L. 80-81: Reproductive season varies depending on the region (between north and southern parts of their distribution)?

We included the following phrases in the introduction for better understanding: “Biological information of *L. santolla* is largely restricted to the southern limit of distribution (Beagle Channel) without data describing reproductive aspects of males derived from their northern distributional limit. Intraspecific latitudinal variability in the reproductive pattern and traits has been documented in crustacean species. Consequently, *L. santolla* in its northern distributional limit (this study) is exposed to different prevailing environmental conditions (e.g. temperature, photoperiod) which possibly have led to variations in the reproductive pattern between the southern and northern limits of its distribution.”.

[Materials and Methods]

L. 107-109: For readers' understanding, provide picture and/or movie from the time of the electroejaculation.

Figure 1a shows a picture of the placement of electrodes during electroejaculation and gonopores where ejaculate is extruded. For further details, methods are described in Soundarapandian (2013) which is cited.

L. 111-113: Describe time periods when authors conducted first and second experiments

The following phrase was added to Material and Methods: “Both temperature conditions
were conducted simultaneously from mid-November to mid-December 2016.”.

[Results]

OK.

[Discussion]

14 L. 275-281: As a result, what method do you think the best to estimate the recovery rate of
15 seminal reserves? Please describe.

We included that following section in the discussion: “We used the two methods to
estimate recovery complementary, as they refer to distinct components of the reproductive
structures. Dry weight of the vasa deferentia represents the total of sperm and seminal fluids,
while the standardized sperm area relates solely to the relative sperm amount. We suggest
that the most appropriate method to be applied in further studies depends on the specific
research question.”.

30 L. 306-320: Delete the conclusion section because this section (L. 307-320) is redundant.

The conclusion section was shorted and now is more concise summarizing only the
most important findings of the study. The conclusion section was not deleted completely to
coincide with the format of the journal Royal Society Open Science.

40 41 **Responses to Reviewer 2**

p2: If the authors want to use the plural of sperm I suggest to use the term "sperms".

Now, the term “sperm” is used in singular. It was corrected in all relevant phrases.

p4: If the authors really meant "physiologically" they have to explain how they have proved
this. But I guess they meant "physically" in terms of body size. The right term is
“physically”.

Indeed we wanted to refer to “physiologically” mature and included the corresponding
citation. “Physiologically” mature refers to individuals which are able to develop gonads.
Southern king crab males reach physiological maturity in the Beagle channel at a size of 75

1
2
3 mm CL. The crabs used in the experiment were ≥ 98 mm CL and therefore can be
considered as “physiologically” mature.

p4: Since the animals are not genetically identical and differ in size and most probably in age
it would be more appropriate in most cases to use the standard deviation.
See also e.g. Altman & Bland 2005 BMJ.

As suggested by the reviewer, instead of using the standard error, we now state the
mean with the standard deviation in the results section and in figure 2.

p4: What is the usual water temperature during mating season?

We indicated the usual water temperature during the mating season and included the
following phrase: “The zone of origin of the individuals of this study is located close to its
northern limit of distribution where crabs reproduce in shallow waters and mean temperatures
present during mating are 9°C (winter mean) and 12°C (annual mean).”

p5: How many slices per section were measured for sperm area?

We added the following phrase to Material and Methods:”One slice of the middle
section of the vasa deferentia, which corresponds to the section where spermatophore
reserves are located, was used to estimate area covered by sperm in the lumen.”

p5: Sperm area should be put in relation to CL and/or diameter of VD.

As suggested by the reviewer, now we present [(sperm area/CL) *1000] to avoid
influence of crab size.

p7: Why is group sample size 4 while in the method parts the authors wrote 5 per group?

Sample sizes are now stated explicitly for each group.

p9: I would say that "brood" has already been fertilized. So better say "eggs".

We thank the reviewer for this observation. “Brood” was changed to “oocytes”.

p9: What about a seasonal aspect in the VDW increase? The authors collected the animals in
October and kept them for a month until the experiments started. The mating season normally
starts in late Nov to early Dec. So maybe there was either some delay in VD filling due to
catching or other artificial factors, or the males of the studied population just usually fill their

VDs later. I think a crucial information would be what temperature is normally present in the
water during mating!

In the Beagle Channel the mating season of *L. santolla* starts in late November to
early December and lasts approximately one month. However, biological information of *L.*
*santolla* is largely restricted to the southern limit of distribution (Beagle Channel) without
data describing reproductive aspects of males derived from their northern distributional limit.
Intraspecific latitudinal variability in the reproductive pattern and traits has been documented
in crustacean species. Consequently, *L. santolla* in its northern distributional limit (this study)
is exposed to different prevailing environmental conditions (e.g. temperature, photoperiod)
which possibly have led to variations in the reproductive pattern between the southern and
northern limits of its distribution, and *L. santolla* in its northern distributional limit possibly
has a different window of reproduction. For this reason we did not conclude a seasonal aspect
of seminal recovery in *L. santolla* in its northern limit of distribution. As requested we
included the usual water temperature during the mating season assuming a larger window of
reproduction of this species but from the northern limit of its distribution and included the
following phrase: "The zone of origin of the individuals of this study is located close to its
northern limit of distribution where crabs reproduce in shallow waters and mean temperatures
present during mating are 9°C (winter mean) and 12°C (annual mean).".

p9: Is natural ejaculation in *L. santolla* an "all or something" action? So, during copulation, is
the whole mature spermatophore packet transferred leaving the VD completely empty? If
eggs of several females could be fertilized by one "load" in subsequent copulations what is
the disadvantage in this? Or what is the point here?

In general, we restructured the discussion and where necessary rephrased for more
clarity. In *L. santolla* the amount of ejaculate delivered during one mating is unknown.
Details about lithodid mating ability are described only in *Paralithodes camtschaticus* and *P.*
*brevipes* which partially transfer sperm of their reserves during one mating resembling a
"something" action. Lithodid crabs are polygamous. The crucial point is that, for example, in
*P. brevipes* an increase in the mating frequency reduces the sperm number ejaculated, the
percentage of spawning females and female fertilization rate. After the second successive
mating of small males (<100 mm CL) partial or null fertilization occurs in females.
Consequently, crustaceans with slow seminal recovery in a male-only fishery which have
mated repeatedly might have depleted seminal reserves especially as the reproductive season
progresses.

p11: I am not sure if I understood this correctly: Mating season for *L. santolla* is during
southern hemispheric summer (starting late Nov to Dec) when water temperature is normally
slightly increased. What is the normal temperature condition during mating in natural
environment (and does mating take place at a certain depth?) for *L. santolla*? I do not really
understand why higher water temperature of 12°C does effect seminal filling/maturing
unfavorable if mating is usually during higher water temperatures (in summer).
At least the author's explanation concerning metabolism is not convincing for me.

Lithodid crabs perform bathymetrical migrations which are associated with
reproduction, where mating takes place in shallow waters. These reproductive migrations
have been observed in the Beagle channel but information is fragmented. The zone of origin
of the individuals of this study is located close to its northern limit of distribution where crabs
reproduce in shallow waters and mean temperatures present during mating are 9°C (winter
mean) and 12°C (annual mean) (see explanation above for likely varying reproductive pattern
between the southern and northern limit of distribution).

*L. santolla* can be fished at sites between 5°C and 12°C along its range of distribution.
The higher water temperature of 12°C likely affects recovery seminal reserves unfavorable
because the origin of the crabs from our study is from its northern distributional limit and
12°C may be close to the threshold of temperature tolerance of *L. santolla*. For example,
deep-water representatives of the Lithodinae sub-family were excluded from waters
exceeding temperatures of 13°C.

We added two reliable references (book section of Willmer et al. 2005 and Paper of
Pörtner 2002) to justify that lower temperature implies a reduction of the standard metabolic
rate (SMR) of the organism and to better explain that therefore more energy may be available
to be assigned to reproduction (faster seminal recovery at 9°C).

p18: N-numbers should be given in all figure legends.

We added N-numbers to all figure legends.

p21: To be honest, this figure is not really informative. At least, areas with sperm should be
marked with a transparent color or in a similar way.

We modified figure 5 for better visibility of sperm areas. Now, figure 5 now shows an
overview of a histological cross-section of the vasa deferentia of *L. santolla* and a detailed
view thereof displaying spermatophores filled with sperm.

Appendix C

Responses to Reviewer Comments

RSOS 181700.R1: Temperature-dependent seminal recovery in the southern king crab *Lithodes santolla*

Katrin Pretterebner, Luis Miguel Pardo, Kurt Paschke

We would like to thank Dr. Jonas Keiler for the revision. We have responded to all of the comments of the reviewer specifically and made appropriate modifications to our manuscript.

Here is the point-by-point list of the modifications:

Reviewer:

2

p9 l 227: The addition of the annual mean is confusing here since mating takes place at 9°C during winter, only. The annual mean does not play a role during mating, does it?

- In the Beagle Channel (southern distributional limit) the mating season of *L. santolla* starts in late November to early December and lasts approximately one month. However, biological information of *L. santolla* is largely restricted to the southern limit of distribution without data describing reproductive aspects of males derived from their northern distributional limit. The exact period of mating of *L. santolla* in its northern distributional limit has not been described yet. Intraspecific latitudinal variability in the reproductive pattern and traits has been documented in crustacean species. Consequently, *L. santolla* in its northern distributional limit (this study) is exposed to different

prevailing environmental conditions (e.g. temperature, photoperiod) which likely have led to variations in the reproductive pattern between the southern and northern limits of its distribution, and *L. santolla* in its northern distributional limit probably has a different window of reproduction. For this reason, we assume an extended window of reproduction in the crab's northern distributional limit using the mean sea surface temperature during the austral winter (9°C) and throughout the year (12°C) as experimental temperatures.

figure page 17: The gonopores are at the coxae!

- In the legend of figure 1a and in the manuscript the following was corrected: gonopores (openings at the coxae of the fifth pereopods).

Additionally, the following remarks in the attached PDF were corrected:

- As suggested by the reviewer we deleted the hypothesis (“decreased metabolism increases seminal production by providing more energy for this process”) and stated the following: “The faster seminal material recovery rate in seawater of 9°C than 12°C after 30 days may be explained by the close proximity of the crabs in this study to their northern distributional limit. Hall and Thatje [44] concluded that deep-water representatives of the Lithodinae sub-family were excluded from waters exceeding temperatures of 13°C. The experimental temperature of 12°C may be close to their threshold of temperature tolerance and therefore, crabs may be energetically less efficient.”.
- According to the suggestion we corrected the following sentence: “Seminal reserves of *L. santolla* were not fully recovered within 30 days after electroejaculation at none of the experimental temperatures

(figure 4).”. New corrected sentence: “Within 30 days after electroejaculation, seminal reserves of *L. santolla* were not fully recovered at any of the experimental temperatures (figure 4).”.

Although the authors have provided the full names for Paralithodes, Callinectes, etc. already in the introduction I would suggest to give the full name if it is mentioned first in the discussion.

- We added the full names of all species mentioned first in the discussion, even if they were already stated in the introduction.

Usage of upper and lower case should be consistent in references.

- We corrected the usage of upper and lower case in the references, which now is consistent.
- As requested, “N” was changed into “n” in all figure legends.
- As requested, exact sample sizes of each sample group was stated in the figure legends and main manuscript.
- We added the following sentence to relate our findings to sustainable fishery: “While our results represent a first step towards improving knowledge on male reproductive traits, we highlight the importance of further research on aspects of reproduction of *L. santolla* in its northern distributional limit to allow a sustainable crustacean fishery in the future.”.
- We corrected small grammatical mistakes and word order as suggested in the PDF, whereby these corrections are not listed here.